# Inflammasome activity is controlled by ZBTB16-dependent SUMOylation of ASC

Danfeng Dong[1,2,12], Yuzhang Du[1,2,12], Xuefeng Fei[1,2,12], Hao Yang[1,2], Xiaofang Li[3], Xiaobao Yang[1,2], Junrui Ma[1,2], Shu Huang[1,2], Zhihui Ma[1,2], Juanjuan Zheng[1,2], David W. Chan [4], Liyun Shi[5], Yunqi Li[6], Aaron T. Irving [7,8], Xiangliang Yuan[1,2], Xiangfan Liu[1,2], Peihua Ni[1,2], Yiqun Hu[1,2], Guangxun Meng [9], Yibing Peng[1,2] ✉, Anthony Sadler[10,11] ✉ & Dakang Xu[1,2] ✉

Inflammasome activity is important for the immune response and is instrumental in numerous clinical conditions. Here we identify a mechanism that modulates the central Caspase-1 and NLR (Nod-like receptor) adaptor protein ASC (apoptosis-associated speck-like protein containing a CARD). We show that the function of ASC in assembling the inflammasome is controlled by its modification with SUMO (small ubiquitin-like modifier) and identify that the nuclear ZBTB16 (zinc-finger and BTB domain-containing protein 16) promotes this SUMOylation. The physiological significance of this activity is demonstrated through the reduction of acute inflammatory pathogenesis caused by a constitutive hyperactive inflammasome by ablating ZBTB16 in a mouse model of Muckle-Wells syndrome. Together our findings identify an further mechanism by which ZBTB16-dependent control of ASC SUMOylation assembles the inflammasome to promote this pro-inflammatory response.

An important element of our immune response is comprised of inflammasome activity. Inflammasome activation results in the assembly of a multi-protein complex initiated by sensors of pathogen-associated or cellular damage-associated molecules with the Apoptosis-associated Speck-like protein containing a CARD (ASC or PYCARD) and pro-Caspase-1. This assembly activates the caspase zymogen, which then cleaves the inactive pro-forms of the inflammatory Interleukin (IL)−1β and −18 cytokines and the pyroptotic Gasdermin D[1,2]. This induces potent and potentially damaging responses and so there is value in determining the processes that control these steps to identify interventions to mitigate pathogenesis caused by excessive inflammasome activity[3,4].

In the resting state, ASC appears in both the nucleus and the cytoplasm. Upon stimulation, it oligomerizes in the cytoplasm to assemble the inflammasome complex[5–8]. We discover that the zinc-finger and BTB domain-containing protein 16 (ZBTB16), also known as

[1]Department of Laboratory Medicine, Ruijin Hospital, Shanghai Jiao Tong University School of Medicine, Shanghai 200025, China. [2]College of Health Sciences and Technology, Shanghai Jiao Tong University School of Medicine, Shanghai, China. [3]Assisted Reproduction Center, Northwest Women's and Children's Hospital, Xi'an, Shaanxi Province 710003, China. [4]School of Medicine, The Chinese University of Hong Kong-Shenzhen, Shenzhen, China. [5]Department of Microbiology and Immunology, Nanjing University of Chinese Medicine, Nanjing, China. [6]Shanghai Institute of Hematology, State Key Laboratory of Medical Genomics, National Research Center for Translational Medicine at Shanghai, Ruijin Hospital, Shanghai Jiao Tong University School of Medicine, Shanghai 200025, China. [7]Department of Clinical Laboratory Studies, Second Affiliated Hospital, Zhejiang University School of Medicine, Hangzhou, China. [8]Centre for Infection, Immunity &Cancer, Zhejiang University-University of Edinburgh Institute, Zhejiang University School of Medicine, Zhejiang University, Haining, China. [9]The Center for Microbes, Development and Health, CAS Key Laboratory of Molecular Virology & Immunology, Shanghai Institute of Immunity and Infection, University of Chinese Academy of Sciences, Shanghai 200031, China. [10]Centre for Innate Immunity and Infectious Diseases, Hudson Institute of Medical Research, 27-31 Wright Street, Clayton, VIC 3168, Australia. [11]Department of Molecular and Translational Science, Monash University, Clayton, VIC, Australia. [12]These authors contributed equally: Danfeng Dong, Yuzhang Du, Xuefeng Fei. ✉e-mail: pyb9861@sina.com; anthony.sadler@hudson.org.au; dakang_xu@163.com

promyelocytic leukemia zinc-finger (PLZF), interacts with ASC in the nucleus to control the conjugation of the Small Ubiquitin-like Modifier (SUMO) to ASC. Post-translational modification of the inflammasome subunits is emerging as a key determinant of activity[9,10]. Although ASC has previously been identified to be phosphorylated and ubiquitylated[11,12], the impact of SUMOylation was unknown.

SUMOylation proceeds by an enzyme cascade initiated by protease maturation by the heterodimeric SAE1/SAE2 (SUMO-activating enzyme subunit-1 and −2) E1 complex followed by covalent transfer, first to the E2 ligase UBC9 (Ubiquitin carrier protein I, also known as Ubiquitin-conjugating enzyme E2 I (UBE2I)) and subsequently to a lysine on the substrate protein[13,14]. The transfer of SUMO may also be shepherded by E3 substrate-recognition enzymes, although this process appears less essential than for the analogous ubiquitination. In the early 2000s, SUMO enzymes and SUMOylated proteins were found in promyelocytic leukemia (PML) bodies[15,16]. These subnuclear structures are assembled by the PML protein and control numerous functions, including immune defence[17]. Although the PML protein is an E3 SUMO enzyme and has been reported to control the cellular location of ASC[7], its SUMOylation is found to be controlled by another constituent of the PML body, ZBTB16.

Both PML and ZBTB16 originally captured attention as translocations with the retinoic acid receptor locus in acute PML patients[18,19]. Reports by us and others have identified that ZBTB16 modulates inflammatory and antiviral immune responses as a transcription factor by directly inducing gene expression via its C-terminal Krüppel-type zinc-fingers and indirectly by recruiting other factors via its N-terminal BTB/POZ and RD2 domains[20–22].

Here we identify a molecular mechanism that controls the central adaptor molecule ASC. Our study identifies that ZBTB16 interacts with ASC, UBC9 and SUMO via its BTB/POZ and RD2 domains. This is shown to facilitate the modification of ASC with one of the three SUMO paralogs, SUMO1[16]. The specific residues on ASC that control its SUMOylation are identified and we demonstrate that the ZBTB16-dependent modification of ASC with SUMO1 in the nucleus promotes the assembly of the inflammasome complex in the cytosol. This deciphers an important molecular mechanism of inflammasome assembly. The impact of this mechanism is tested in cell lines and in vivo with specific inflammatory stimuli using constitutive and targeted Zbtb16 mutant animals as well as a mouse model of Muckle-Wells syndrome that has a hyperactive inflammasome. Together these data identify a ZBTB16-SUMO1-ASC axis that promotes inflammasome activity with consequences for inflammatory immunity.

## Results

### ZBTB16 promotes inflammasome activity

We previously reported that ZBTB16 repressed Nuclear factor (NF)-κB-mediated gene induction by stabilising the nonproductive NF-κB1 (p50) transcription factor on gene promoters[21,22]. Considering the role of NF-κB in the expression of IL-1β and Gasdermin D (GSDMD), we tested the physiological consequence of ZBTB16 for inflammasome activity by challenging WT and Zbtb16 mutant mice with a variety of inflammasome activators.

We first tested a model of acute peritonitis induced by intraperitoneal injection of monosodium urate (MSU) crystals. MSU acts as a universal damage-associated molecular pattern and is responsible for synovial inflammation in gout patients by activating Nod-like receptor (NLR) pyrin containing 3 (NLRP3)[23]. Responses were compared between WT mice and: Whole-body Zbtb16 knockout mice (Zbtb16[-/-]); myeloid cell-specific Zbtb16 knockout mice that had been engineered with Cre recombinase recognition sites (Zbtb16[fl/fl]) on a myelomonocytic cell-specific lysozyme M (LysM) background (Zbtb16[fl/fl]LysM[Cre]) (Supplementary Fig. 1) or mice ablated for ASC (Asc[-/-]) as a non-responsive control. Expectedly, ablating ASC decreased the response to MSU compared with WT mice, as measured by the morphology of

the peritoneum and scoring of the inflammatory index (Fig. 1a, b), the recruitment of neutrophils to the peritoneal cavity (Fig. 1c and d) and the levels of IL-1β in peritoneal fluid (Fig. 1e). Unexpectedly, ablating Zbtb16 expression in the whole animal or merely in the myeloid cell lineage induced a similar immune impairment as ablating ASC (Fig. 1a−e). In contrast, there was no difference in the levels of inflammasome-independent IL-6 in the peritoneal fluid between WT and Zbtb16[-/-] mice, indicating a degree of specificity in this response (Fig. 1f).

We next examined the response of BMDMs isolated from WT or Zbtb16[-/-] mice to different inflammasome triggers. ELISA and an immunoblotting assay of BMDMs stimulated with MSU and a range of other NLRP3 activators confirmed the defect observed in vivo as reduced processing of pro-IL-1β, pro-IL-18 and pro-Caspase-1 in the Zbtb16[-/-] cells (Fig. 2a−d). Notably, the expression of NLRP3, ASC, GSDMD and pro-IL-1β as well as the NIMA-related kinase 7 (NEK7), which is essential for NLRP3 inflammasome activation[24,25], were unaffected by Zbtb16 expression (Fig. 2c, d and Supplementary Fig. 2a, b).

To trial an inflammasome trigger that is less dependent upon NLRP3, we tested the response to the microbe C. difficile. C. difficile infection induces severe colitis due, in part, to the effects of the microbial glucosyltransferase toxin B (TcdB) on ASC function[26,27]. Measures of the production of IL-1β by BMDMs or macrophages isolated from the peritoneal cavity in response to either cultured C. difficile or the purified C. difficile TcdB toxin showed that ablating Zbtb16 impaired inflammasome activity (Fig. 2e). The loss of Zbtb16 also resulted in decreased cytotoxicity, as indicated by the release of LDH in response to C. difficile, its TcdB toxin or nigericin (Fig. 2f).

This activity of ZBTB16 was alternatively tested by acutely knocking out the gene in the human monocytic THP-1 cell line using DOX-inducible CRISPR gene editing. The phenotype of these cells was validated by the detection of fluorescent reporters for Cas9 and guide RNA expression, measures of the ZBTB16 transcript and rescue of the DOX-induced deficiency by exogenous re-expression of ZBTB16 (Supplementary Fig. 2c−e). Transiently decreasing ZBTB16 in these cells reduced IL-1β release and cytotoxicity in response to treatment with LPS and nigericin or by treatment with C. difficile or its purified TcdB toxin (Fig. 2g).

Together these data identify that ZBTB16 affects inflammasome activity without any conspicuous effect on the levels of the signalling components. Accordingly, ZBTB16 does not appear to be controlling inflammasome activity by its recognised function as a transcription factor.

### ZBTB16 acts downstream of inflammasome sensors

To assess the physiological impact on inflammasome activity we tested the effect of ZBTB16 against the hyper-activation of this pathway. Mice carrying a knock-in mutation of Nlrp3 (R258W), which corresponds to a causal mutation in cryopyrin-associated periodic syndrome patients[28], were crossed with Zbtb16[-/-] mice to generate animals expressing hyperactive NLRP3 in normal or Zbtb16-deficient backgrounds. Remarkably, the chronically elevated level of IL-1β in the serum of Nlrp3[R258W] neonates was greatly attenuated in their Nlrp3[R258W]/Zbtb16[-/-] littermates (Fig. 3a). This reduction in the production of the cytokine suppressed the severity of spontaneous skin inflammation in the posterior collar area and perianal region of the Nlrp3[R258W]/Zbtb16[-/-] mice compared to their Nlrp3[R258W] littermates (Fig. 3b and Supplementary Fig. 3).

The previously established hypersensitivity of the Nlrp3[R258W] mice was tested by delayed-type hypersensitivity (DTH) induction with DNCB[28]. The hyperkeratosis and a neutrophil infiltrate, with the related MPO release, that is induced in the skin of Nlrp3[R258W] mice with this treatment was significantly attenuated by ablating Zbtb16 (Fig. 3c, d). The fact that ablating Zbtb16 ameliorates the effects of a hyperactive Nlrp3 mutant demonstrates the impact of ZBTB16 on inflammasome activity. The effectiveness of ZBTB16 to moderate inflammation in the

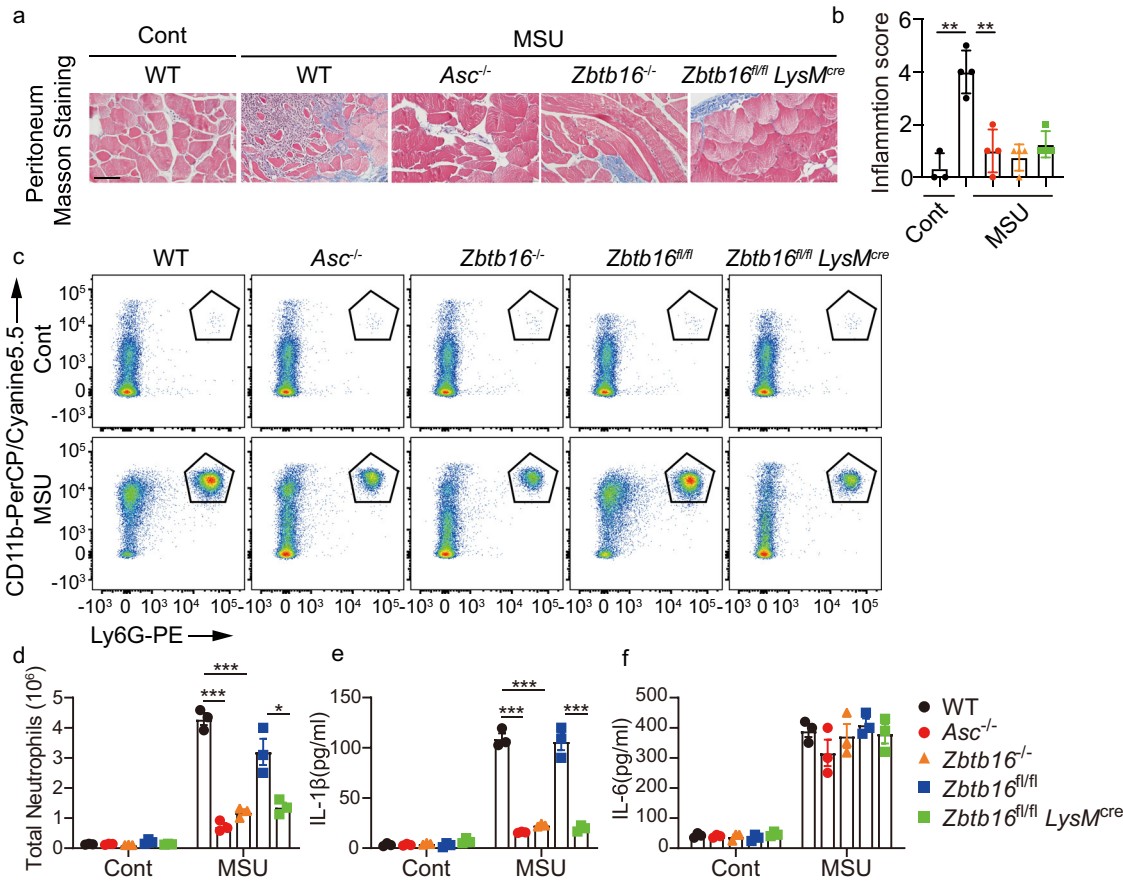

**Fig. 1 | ZBTB16 promotes inflammasome-mediated pathogenesis.** WT, *Asc*[-/-], *Zbtb16*[fl/fl] and *Zbtb16*[fl/fl]LysM[Cre] mice were intraperitoneally injected with 3 mg monosodium urate (MSU) crystals and then assessed by (**a**) histological analysis of peritoneum from the indicated strains exposed to MSU processed by Masson staining to visualise pathogenesis (the scale bar = 50 µm) and (**b**) by an inflammation score, calculated as the sum of neo angiogenesis (0 when <3, 1 when 4–8, 2 when 9–12, 3 when >12 vessels) with polymorphonuclear cell (PMN) accumulation at the site of the injury (0 when <3, 1 when 4–8, 2 when 9–12, 3 when >12 activated PMNs are present) (*n* = 3 for control and *n* = 4 for MSU treated mice per group). After 6 h exposure to MSU peritoneal cells were collected from mice with 500 µl PBS lavage and assessed by counting the numbers of neutrophils (as CD11b⁺Ly6G⁺) by flow cytometry. The data are displayed as (**c**) representative dot plots and (**d**) a count of the total number of neutrophils in each condition. The levels of the inflammatory cytokines (**e**) IL-1β and (**f**) IL-6 in peritoneal fluids as measured by ELISA are shown as means ± SEM (*n* = 3 mice per group in each experiment). Two-tailed Student's *t*-test were calculated for; WT vs *Asc*[-/-] $p = 8.6 \times 10^{-5}$ or $5.5 \times 10^{-5}$, WT vs *Zbtb16*[-/-] $p = 1.4 \times 10^{-4}$ or $7.8 \times 10^{-5}$ or *Zbtb16*[fl/fl] vs *Zbtb16*[fl/fl]LysM[Cre] $p = 1.6 \times 10^{-2}$ or $6.3 \times 10^{-4}$ for the comparison of neutrophils or IL-1β, respectively (**$p < 0.01$ or ***$p < 0.001$). Source data are provided as a Source Data file.

face of constitutive NLRP3 activity also suggests that ZBTB16 functions downstream of the inflammasome sensor proteins.

## ZBTB16 regulates the formation of the inflammasome complex

Activation of inflammasome sensors triggers the oligomerization of the ASC adaptor protein. Therefore, we next investigated whether Zbtb16 deficiency affected ASC oligomerization in BMDMs upon stimulation with inflammasome activators. Towards this, cells were treated with LPS alone or LPS and the NLRP3 activator nigericin, followed by Triton X-100 then cross-linked with disuccinimidyl suberate and analysed by immune-blotting with an anti-ASC antibody. Figure 4 shows that ASC oligomerization, determined as higher molecular weight fractions and Triton X-100 insoluble ASC after treatment with nigericin, was reduced by ablating *Zbtb16* (Fig. 4a). Equivalently, quantification of the formation of ASC specks in the cytosol of BMDMs by immunofluorescence shows that ablating *Zbtb16* reduced ASC oligomerization in response to nigericin, adenosine triphosphate (ATP) and *C. difficile* or its TcdB toxin (Fig. 4b). *Zbtb16* did not alter ASC expression, as measured by immunoblot and immunofluorescence (shown previously in Fig. 2c, d and in this experiment in Supplementary Fig. 4).

As ASC oligomerization induces the processing of the Caspase-1 zymogen, we measured its activity. A fluorescent Caspase-1 assay (FLICA[29]) detected heightened Caspase-1 activity in the WT compared to the *Zbtb16*[-/-] BMDMs treated with LPS and nigericin (Fig. 4c). The subsequent processing of cytokines by active Caspase-1 requires their recruitment to the inflammasome. We visualised the recruitment of a pro-IL-1β construct that was tagged at its termini with fluorophores as described previously[30]. This shows the increased aggregation of IL-1β into cytosolic puncta in the WT compared to the *Zbtb16*[-/-] BMDM treated with LPS and nigericin (Fig. 4d). Further evidence for ZBTB16-dependent assembly of the inflammasome was demonstrated by measuring the interaction between ASC and NLRP3 or Caspase-1 by immunoblotting of immunoprecipitants from the lysates of WT and *Zbtb16*[-/-] cells treated with nigericin (Fig. 4e).

These data identify that ZBTB16 promotes ASC oligomerization and the subsequent assembly of the active inflammasome complex.

## ZBTB16 promotes ASC SUMOylation

Previous reports identified that the E3 SUMO ligase PML, which co-locates with ZBTB16, regulates ASC nuclear location[31,32]. As PML-nuclear bodies are SUMOylation hotspots, with ZBTB16 itself a SUMO

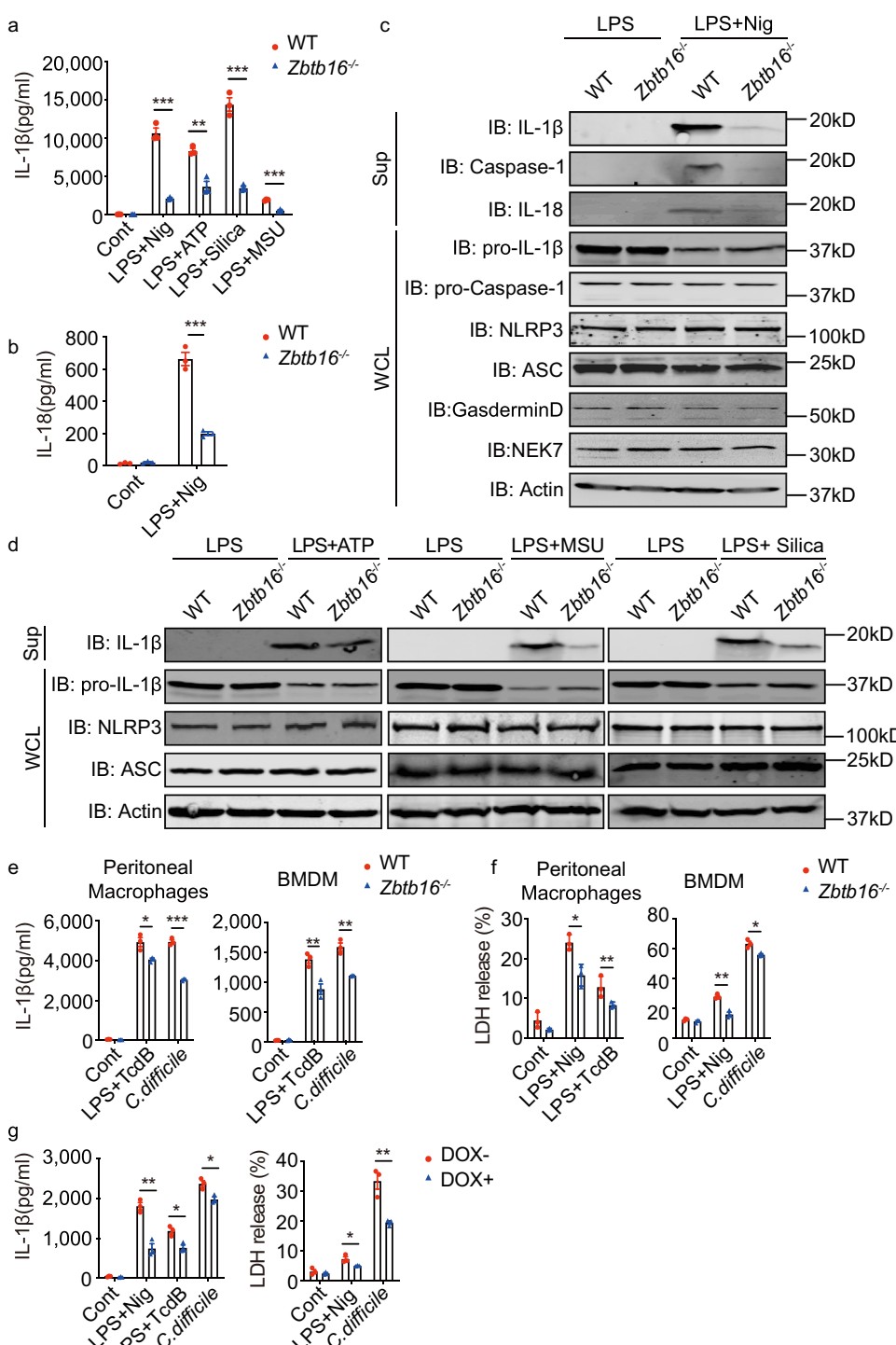

**Fig. 2 | ZBTB16 promotes inflammasome activation.** BMDMs from WT and *Zbtb16*[-/-] mice were either left untreated or primed with 1 μg/mL LPS for 4 h, followed by stimulation with 20 μM nigericin (Nig) or 5 mM ATP for 30 min, 120 μg/mL silica or 200 μg/mL MSU for 6 h, then the levels of the (**a**) IL-1β and (**b**) IL-18 cytokines were analysed in the supernatants by ELISA (*n* = 3, Data are presented as mean values ± SD) and (**c, d**) the cell lysates and supernatants were immunoblotted (IB) with the indicated antibodies. **e** IL-1β production by WT and *Zbtb16*[-/-] mouse peritoneal macrophages and BMDMs were primed with 1 μg/mL LPS for 2 h then stimulated with 200 ng/mL of TcdB toxin for 4 h or *C. difficile* 630 at MOI 100 for 6 h.

**f** Cytotoxicity was assessed by LDH release in peritoneal macrophages and BMDMs treated with LPS then TcdB or 20 μM nigericin, or treated with *C. difficile* 630 for 30 min. **g** IL-1β production by THP-1 cells expressing ZBTB16 (DOX-) or ablated for ZBTB16 (DOX + , 1 μg/mL for 96 h) followed by treated with LPS and TcdB or nigericin or, alternatively, treated with *C. difficile* 630 (*n* = 3, data are presented as mean values ± SD for **e**–**g**) was measured by ELISA in cell supernatants. Data are from at least three independent experiments. Statistical differences (*$p < 0.05$, **$p < 0.01$ or ***$p < 0.001$) were determined by a two-tailed Student's *t*-test. Source data are provided as a Source Data file.

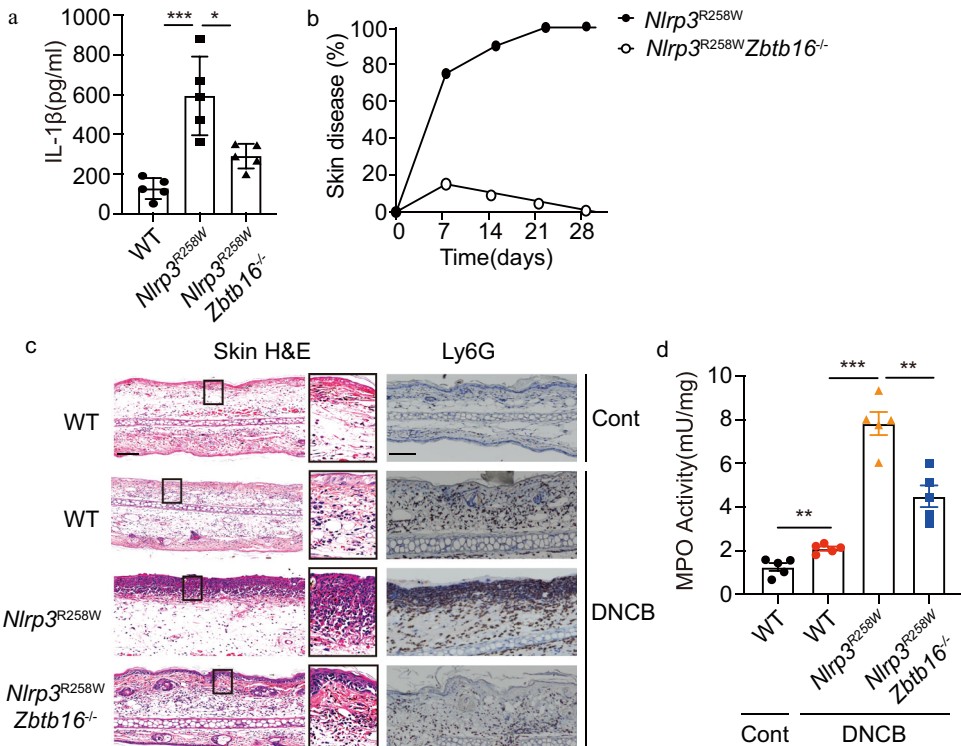

**Fig. 3 | ZBTB16 controls a gain-of-function mutant inflammasome. a** The serum levels of IL-1β were examined from WT, *Nlrp3*^R258W and *Nlrp3*^R258W*Zbtb16*^-/- mice. **b** The percentage of *Nlrp3*^R258W and *Nlrp3*^R258W*Zbtb16*^-/- mice that developed skin inflammation within one month of birth is shown (*n* = 20). **c, d** 1-chloro-2,4-dinitrobenzene (DNCB) -induced skin inflammation in the indicated mice. **c** Representative H&E-stained sections of skin tissues showing the epidermal thickness and infiltration of neutrophils, as assessed by immunohistochemical

staining with an anti-Ly6G antibody (scale bar = 75 μm) and by (**d**) measures of MPO activity as quantified by ELISA in skin extracts from untreated controls or 6 days after DNCB treatment. Error bars represent the mean ± SEM of technical replicates (*n* = 5 mice per group in each experiment). Statistical differences (**\**p* < 0.01 or ***\**p* < 0.001) were determined by a two-tailed Student's *t*-test. Source data are provided as a Source Data file.

substrate, we tested if ASC was modified with SUMO. To directly assess the SUMOylation of ASC, we transfected HEK-293T cells with ASC, the SUMO ligase UBC9 and His-tagged constructs of SUMO1-3 then enriched SUMO and its conjugates by binding to Nickel-NTA agarose resin (Ni-NTA) as described previously[33]. This identified that ASC is modified by SUMO1 (Fig. 5a and Supplementary Fig. 5a). Fittingly, this SUMOylation of ASC was reduced by either treatment with a pharmacological inhibitor (2-D08) or co-expression of the SUMO1-specific protease Sentrin-specific protease 1 (SENP1) (Fig. 5b, c)[34]. To evaluate the role of ZBTB16 in ASC SUMOylation, we overexpressed ASC, His-SUMO1 and UBC9 with ZBTB16 in HEK-293T cells. Immunoblotting of Ni-NTA enriched cell lysates detects ZBTB16 expression increased ASC SUMOylation (Fig. 5d). This ZBTB16-dependent increased ASC SUMOylation was confirmed for the endogenous proteins by immunoblotting peptides enriched with an anti-ASC or control anti-IgG antibodies from WT and *Zbtb16*^-/- BMDMs with an anti-SUMO1 antibody (Fig. 5e). Consistent with these data, ASC and SUMO1 were detected to colocalize to a greater extent in the nucleus of WT compared to *Zbtb16*^-/- BMDMs after stimulation with LPS and nigericin as measured by immunofluorescence (Fig. 5f). Notably, Zbtb16 did not alter the expression of the components of the SUMO pathway (Supplementary Fig. 5b, c).

These results identify that ASC is post-translationally modified by SUMO1 and that this SUMOylation is controlled by ZBTB16 in the cell nucleus.

## ZBTB16 configures an ASC SUMOylation complex
To assess if ZBTB16 might directly control the modification of ASC we assessed the association between the proteins. Measures of the localisation of the SUMO factors were carried out.

Immunohistochemical labelling of BMDM using antibodies against ASC, SUMO1, the SUMO ligase UBC9, ZBTB16 and PML reveals that the proteins predominantly localise in the nucleus of unstimulated cells (Fig. 5f and Supplementary Fig. 6). Analysis of the expression of ASC and ZBTB16 detected by immunohistochemical double-labelling of BMDM transformed with a ZBTB16 expressing construct detects a degree of colocalization of ZBTB16 and ASC in the nucleus in resting cells. The corresponding reconstructed renderings by Imaris showed the contact area of ZBTB16 with ASC (5.45 μm²) (Fig. 6a and Supplementary Fig. 6). An association between ASC and ZBTB16 was supported by the co-immunoprecipitation of the proteins when overexpressed in HEK-293T cells (Fig. 6b, c). To detect if these proteins interact in situ at endogenous levels, we performed fluorescent proximity ligation assays (PLA) for the association of ASC with ZBTB16, SUMO1 or NLRP3 in BMDM cells that were untreated or treated with LPS alone or in combination with nigericin. A fluorescence signal was generated between antibodies for ASC and ZBTB16 in the nucleus of resting WT BMDMs, but not the *Zbtb16*^-/- cells, which increased after immune stimulation (Fig. 6d). This protocol also confirmed the previously detected immune-mediated association between ASC and SUMO1 (Figs. 5f and 6d). Critically, this signal was heightened in the WT compared with the *Zbtb16*^-/- BMDM, thereby supporting a positive role for ZBTB16 in ASC SUMOylation (Fig. 6d). ZBTB16 was also shown to promote the assembly of the inflammasome (shown in Fig. 4), as apparent from the heightened association between ASC and NLRP3 upon immune stimulation in the WT compared to the *Zbtb16*^-/- BMDM (Fig. 6d). This assay confirms that ZBTB16 colocalizes with ASC to increase its colocalization with SUMO1 in the nucleus and subsequently promotes an association with NLRP3 in the cytosol.

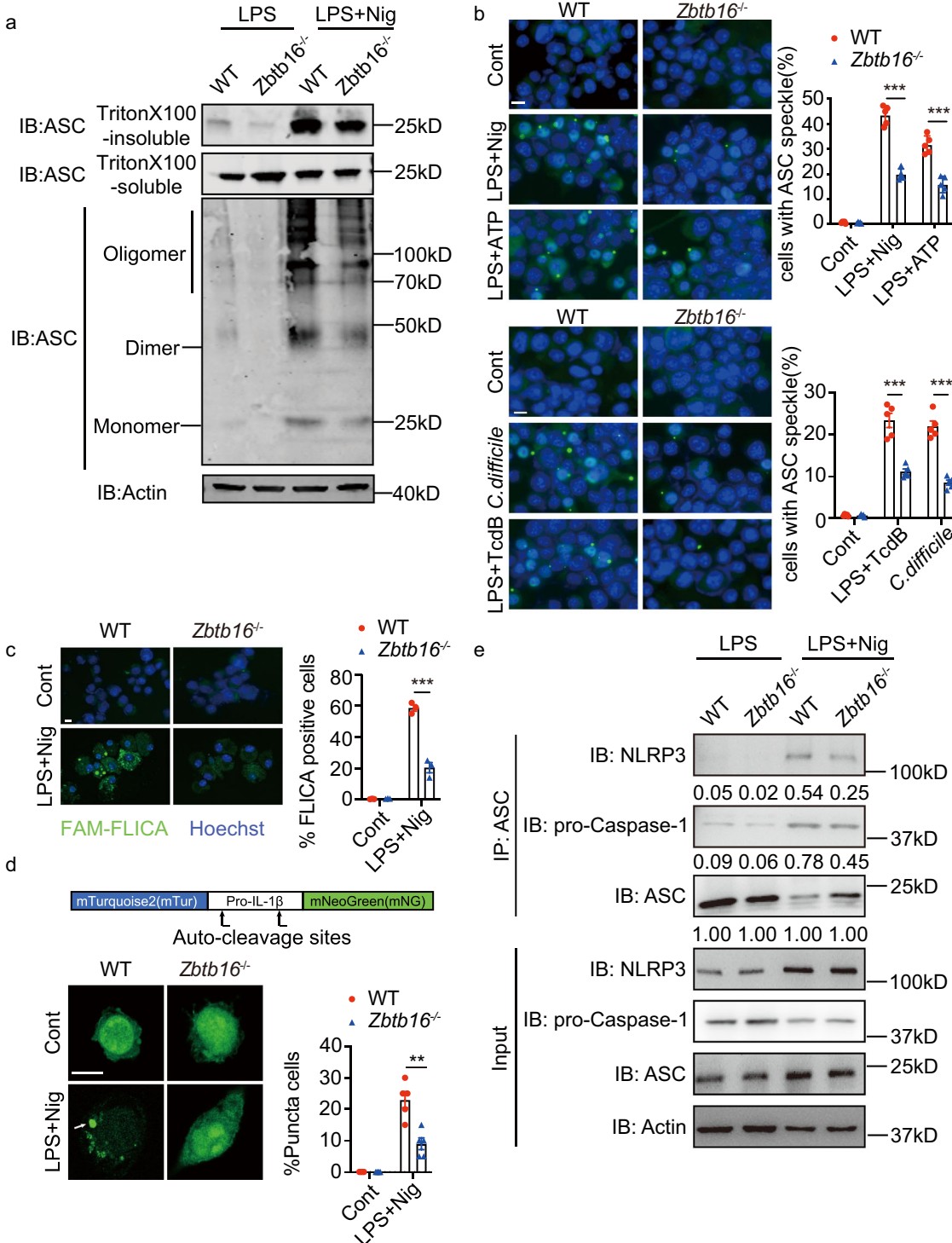

**Fig. 4 | ZBTB16 promotes ASC oligomerization. a** BMDMs from WT and *Zbtb16*[-/-] mice were primed with 100 ng/mL LPS for 4 h and stimulated with nigericin followed by chemical cross-linking and centrifugation, then the pellets (Triton X 100-insoluble) and supernatants (Triton X 100-soluble) were probed by IB with an anti-ASC antibody. **b** Immunofluorescent detection of ASC specks in control untreated or BMDMs primed with LPS then stimulated with nigericin, ATP or TcdB, or treated with *C. difficile* alone and then stained with an anti-ASC antibody (green) and DAPI (blue) to visualised perinuclear ASC specks (scale bar = 10 μm) (*n* = 5). Representative micrographs are shown on the left and ASC specks in the cell cytosol are quantified on the right. **c** Caspase-1 activity in BMDMs primed with LPS and stimulated with nigericin is measured by FLICA assay (FAM-YVAD-FMK). Representative micrographs are shown on the left and quantification of FLICA-positive cells is graphed on the right as determined for 250 cells (scale bar = 10 μm) (*n* = 3). **d** Measures of the aggregation of a fluorescent-tagged pro-IL-1β as a cytosolic speck

in BMDMs from WT and *Zbtb16*[-/-] mice transfected with the construct, then 24 h later stimulated with LPS and nigericin. The fluorescent signal in representative cells is shown on the left and the proportion of cells with GFP specks is quantified from 20 fluorescent positive cells in each field (scale bar = 10 μm). The percentage of cells with fluorescent puncta was scored in the graph on the right (*n* = 5). Data are presented as mean values ± SEM for (**b**–**d**). **e** An assessment of the association of Asc with Nlrp3 and Caspase-1 by immunoprecipitation of lysates from treated BMDMs with an anti-ASC antibody then probing with the indicated fluorescently tagged antibodies. Quantification of each protein by their relative fluorescence in the IB is shown below the detected bands. All data are representative of at least three independent biological experiments. Statistical differences (**p < 0.01 or ***p < 0.001) were determined by a two-tailed Student's *t*-test. Source data are provided as a Source Data file.

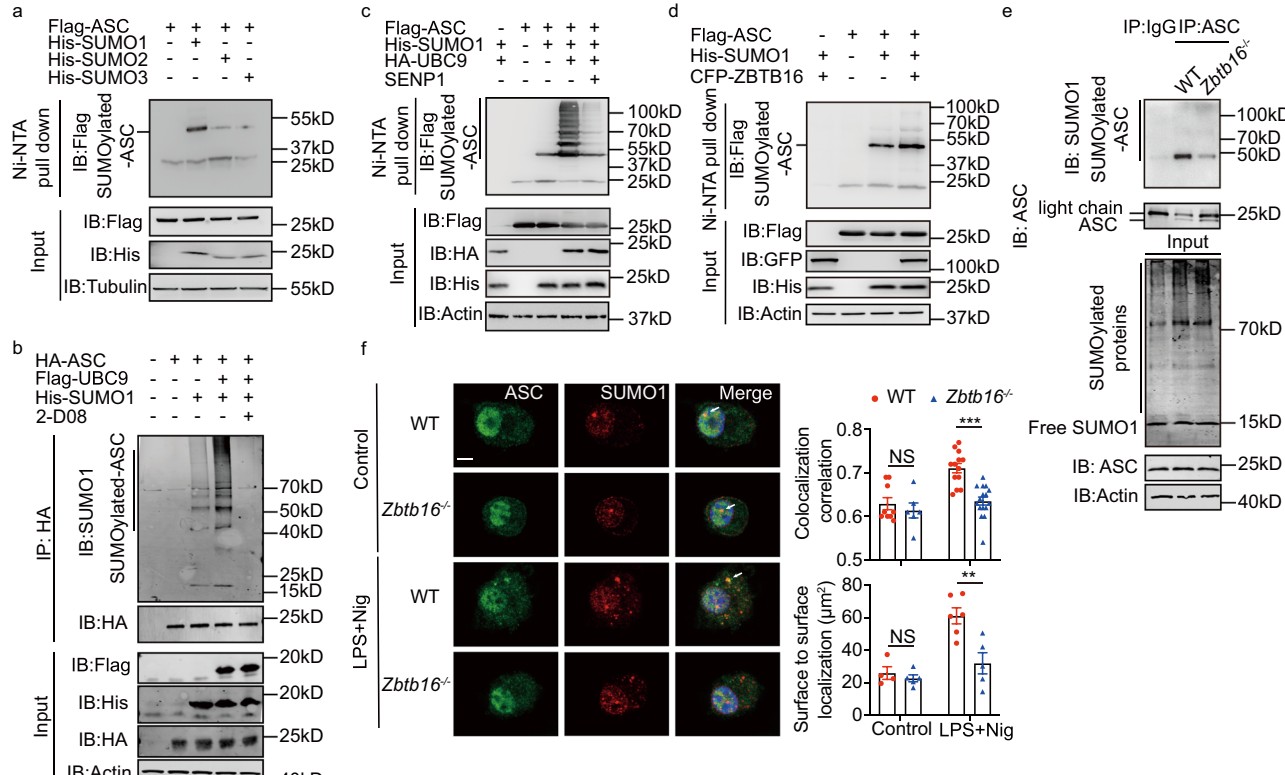

**Fig. 5 | ZBTB16 promotes ASC SUMOylation. a** Detection of ASC SUMOylation by IB of lysates from HEK-293T cells co-transfected with HA-UBC9, Flag-ASC and His-SUMO1, His-SUMO2 or His-SUMO3, followed by precipitation with Ni-NTA resin to enrich SUMOylated proteins then probing with the indicated antibodies. Confirmation of the conjugation of SUMO1 to ASC by IB of lysates from HEK-293T cells co-transfected with tagged UBC9, ASC and SUMO1 and either (**b**) treated with the SUMOylation inhibitor 2-D08 (100 μM) enriched by HA beads or (**c**) by expression of the SUMO1 protease SENP1. SUMOylated proteins were enriched with Ni-NTA resin and then probed with the indicated antibodies. **d** Detection of ZBTB16-dependent promotion of ASC SUMOylation by IB of lysates from HEK-293T cells expressing CFP-ZBTB16, His-SUMO1 and Flag-ASC enrichment with Ni-NTA resin then probed with the indicated antibodies. **e** A confirmation that endogenous ASC SUMOylation is promoted by ZBTB16 in BMDM cells by IP of SUMOylated proteins from WT and *Zbtb16*[-/-] BMDM lysates with an anti-ASC antibody and then IB with an

anti-SUMO1 antibody. **f** Measures of the effect of ZBTB16 on the colocalization of ASC and SUMO1 in WT and *Zbtb16*[-/-] BMDM cells untreated (Control) or primed with LPS for 4 h then treated with nigericin (LPS+Nig) to activate the Nlrp3 inflammasome. Representative micrographs showing single-plane confocal images of WT and *Zbtb16*[-/-] BMDMs cells with anti-ASC (green) and anti-SUMO1 (red) antibodies (scale bars = 10 μm). The extent of the colocalization of the ASC and SUMO1 signals is quantified in the graphs on the right with the IMARIS software (*n* = 9 for WT control, *n* = 6 for *Zbtb16*[-/-] control, *n* = 13 for WT stimulated and *n* = 16 for *Zbtb16*[-/-] stimulated, data are presented as mean values ± SD). Colocalization correlations and surface-to-surface localisation was determined in randomly selected cells over five fields. Each data point represented the mean value in the field. All data are representative of at least three independent experiments. Statistical differences (***\*\*\*p* < 0.001) were determined by a two-tailed Student's *t*-test. Source data are provided as a Source Data file.

We next sought to map the interacting domains of the protein partners. For this, we used a bimolecular fluorescence complementation assay. This assay evaluates protein associations by tagging partners with the separate amino- or carboxyl-terminal fragments of a split-fluorophore. An association between the separate protein partners reconstitutes the full-length fluorophore and so is detected as a fluorescent signal[35,36]. ZBTB16, ASC, UBC9 and SUMO1 were differently tagged with the separate halves of split-Venus (coded V1 and V2) and co-expressed in HEK-293 cells. As well as the full-length ZBTB16, truncated constructs encoding the proteins' Broad-Complex, Tramtrack and Bric-a-brac (BTB) domain alone and this with the adjacent repressor domain (BTB-RD) were also tested. In addition, SUMO1 was alternatively tagged at either terminus to distinguish merely an association from covalent conjugation, based upon amino-tagging leaving free while carboxyl-tagging blocks the carboxyl diglycine residues required for SUMOylation. The pattern of Venus fluorescence produced by the oligomerization of ASC and its association with UBC9 or SUMO1 in HEK293 cells is shown (Fig. 6e). Two distinct patterns of fluorescence were produced by the ASC associations, with multiple nuclear speckles evident for the heterogeneous pairings and a single cytosolic speck produced by ASC oligomerization (Fig. 6e). Measures of the total level of Venus fluorescence in cells transfected with the

different ZBTB16 constructs and either ASC or UBC9 confirm their association and distinguish that they separately associate with the BTB and repressor domains of ZBTB16, respectively (Fig. 6f). Fluorescence produced by an association between ZBTB16 and the N-terminally tagged SUMO1 (nSUMO1) was partly retained with the carboxyl-terminally tagged SUMO1 (SUMO1c), thereby identifying that ZBTB16 can colocalize with SUMO1 other than as a substrate[37]. Importantly, although the colocalization between ZBTB16 and ASC or UBC9 produced low levels of Venus fluorescence compared to that of ASC and UBC9 or SUMO1, the expression of ZBTB16 significantly increased the level of fluorescence produced between tagged ASC and SUMO1, thereby supporting a positive function of ZBTB16 in the association of ASC with SUMO1 (Fig. 6g).

This with the preceding data is consistent with ZBTB16 functioning to configure the substrate and the E2 enzyme to promote ASC SUMOylation in the nucleus.

## Lysine residues 21 and 109 direct ASC's SUMOylation and oligomerization

To identify the residues that are modified by SUMO1 we conducted arginine replacement mutagenesis of lysines on ASC and co-expressed each as a Flag-tagged construct with SUMO1 and UBC9 in HEK-293T

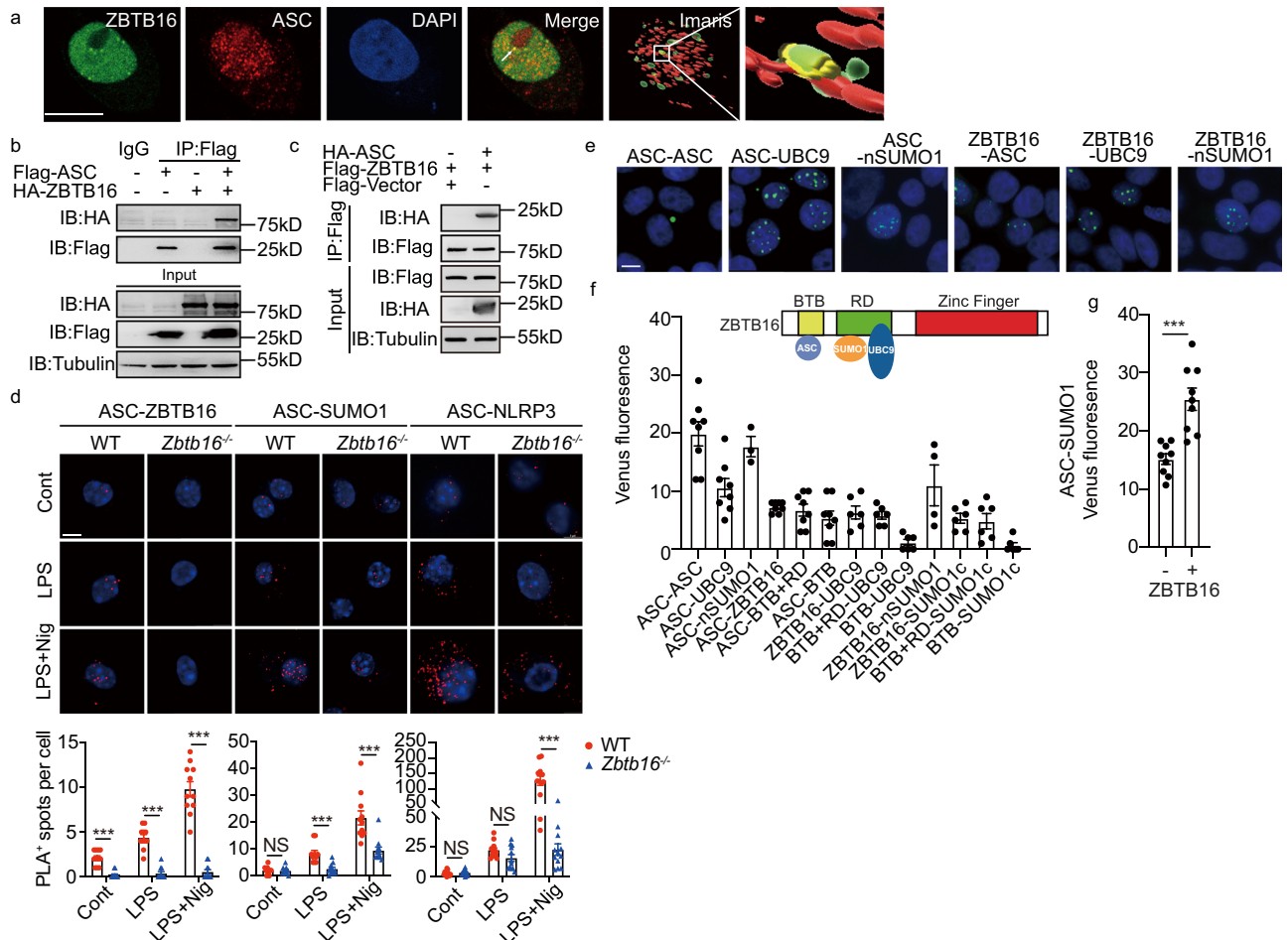

**Fig. 6 | ZBTB16 configures an ASC SUMOylation complex. a** Z-stack micrographs showing the nuclear location of ZBTB16 and ASC in BMDMs by immuno-fluorescence in cells transfecting with ZBTB16 and stained with DAPI (blue), mouse anti-ZBTB16 and rabbit anti-ASC primary antibodies followed by Alexa Fluor 488 anti-mouse (green) and Alexa Fluor 555 anti-rabbit (red) secondary antibodies. The corresponding reconstructed renderings by Imaris 9.5 (rightmost) for quantification of the contact area of ZBTB16 with ASC. Yellow indicated contact surface (scale bar = 10 μm). **b, c** Detection of an association between ZBTB16 and ASC by IB of lysates from HEK-293T cells co-transfected with tagged ASC and ZBTB16, immu-noprecipitated (IP) with anti-Flag antibodies then probed with the indicated anti-bodies. **d** Representative fluorescent micrographs of WT and *Zbtb16*−/− BMDM cells stained by Proximity ligation assay (PLA) with antibodies for ASC and ZBTB16, SUMO1 or NLRP3. The cells were untreated (Cont) or primed with LPS for 4 h (LPS) and then stimulated with nigericin (LPS+Nig). The fluorescent signal is quantified in the graphs (below) as the mean ± s.e.m. Results are shown for three independent experiments with each data point representing 11–15 cells taken from 5 different views for each group. Cell nuclei are stained blue by DAPI (scale bar = 10 μm). **e** Fluorescent micrographs of HEK-293 cells expressing the indicated pairs of the following split-Venus tagged constructs: V1-ASC, V2-ASC, UBC9-V2, V1-SUMO1

(nSUMO1), SUMO1-V1 (SUMO1c), full-length V2-ZBTB16, ZBTB16 with the Zinc-finger domain removed (V2-BTB-RD) and just the BTB domain of ZBTB16 (V2-BTB). Cell nuclei are visualised by Hoechst staining (blue). As different exposure times were used to capture the separate images, the relative level of fluorescence is not representative (scale bars = 10 μm). **f** Total Venus fluorescence in HEK-293 cells co-transfected with the indicated split-Venus constructs and normalised against a negative control. The variance shows the standard error of the means from inde-pendent experiments (n = 8 for V1-ASC + V2-ASC, V1-UBC9 + V2-ASC, V1-ASC + V2-ZBTB16 / -RD-BTB / -BTB; n = 6 for V1-UBC9 + V2-ZBTB16 / -BTB-RD / -BTB, SUMO1-V1 + V2-ZBTB16 / -BTB-RD / -BTB; n = 4 for V1-SUMO1 + V2-ZBTB16; n = 3 for V1-SUMO1 + V2-ASC). The inset schematic depicts the predicted association of ASC, UBC9 and SUMO1 with the different protein domains of ZBTB16 by the bimolecular complementation. **g** A graph quantifying the relative levels of Venus fluorescence generated by an association between V1-SUMO1 and V2-ASC in HEK-293 cells co-transfected with constructs expressing UBC9 and either ZBTB16 or, as a control, the empty backbone vector (n = 9 per group, data are presented as mean ± SD). Sta-tistical differences (***p < 0.001) were determined by a two-tailed Student's t-test. Source data are provided as a Source Data file.

cells, then assessed SUMOylation as previously described. As this experiment did not reveal a change in ASC SUMOylation despite separately mutating every lysine, and because the Flag epitope con-forms to a SUMO motif (Supplementary Fig. 7a–c), we repeated the experiment with an HA-tagged ASC. These constructs identify that the lysine residues at positions 21 and 109 impacted ASC SUMOylation (Fig. 7a, b). Computer-aided prediction (by SUMOplot) distinguishes these same residues as putative SUMOylation sites (Fig. 7c). The lysine residue at position 109 is part of a flexible link between the proteins caspase and recruitment domain (CARD) and pyrin domain (PYD), while the lysine at position 21 forms a pocket with lysine residues number 22, 24 and 26 on the PYD of ASC (Fig. 7c)[38]. We exploited the

Flag tag as an affixed SUMO site to try to distinguish how these two regions affected ASC SUMOylation. Mutation of the lysine residues between positions 21 to 26 (coded 4KR) reduced the SUMOylation of the Flag-ASC construct (Supplementary Fig. 7d, e). This curtailed SUMOylation, even at extrinsic sites in the Flag epitope, suggests that the residues between 21 and 26 affect the recruitment of ASC to the SUMOylation complex. In keeping with this, mutating the lysine resi-dues 21−26 affected ASC's association with ZBTB16, as assessed by immunoblot analysis of WT and the 4KR mutant ASC proteins over-expressed and immunoprecipitated with ZBTB16 in HEK-293T cells (Fig. 7d). Accordingly, this mutation also ablated the ZBTB16-dependent increase in ASC SUMOylation (Fig. 7e).

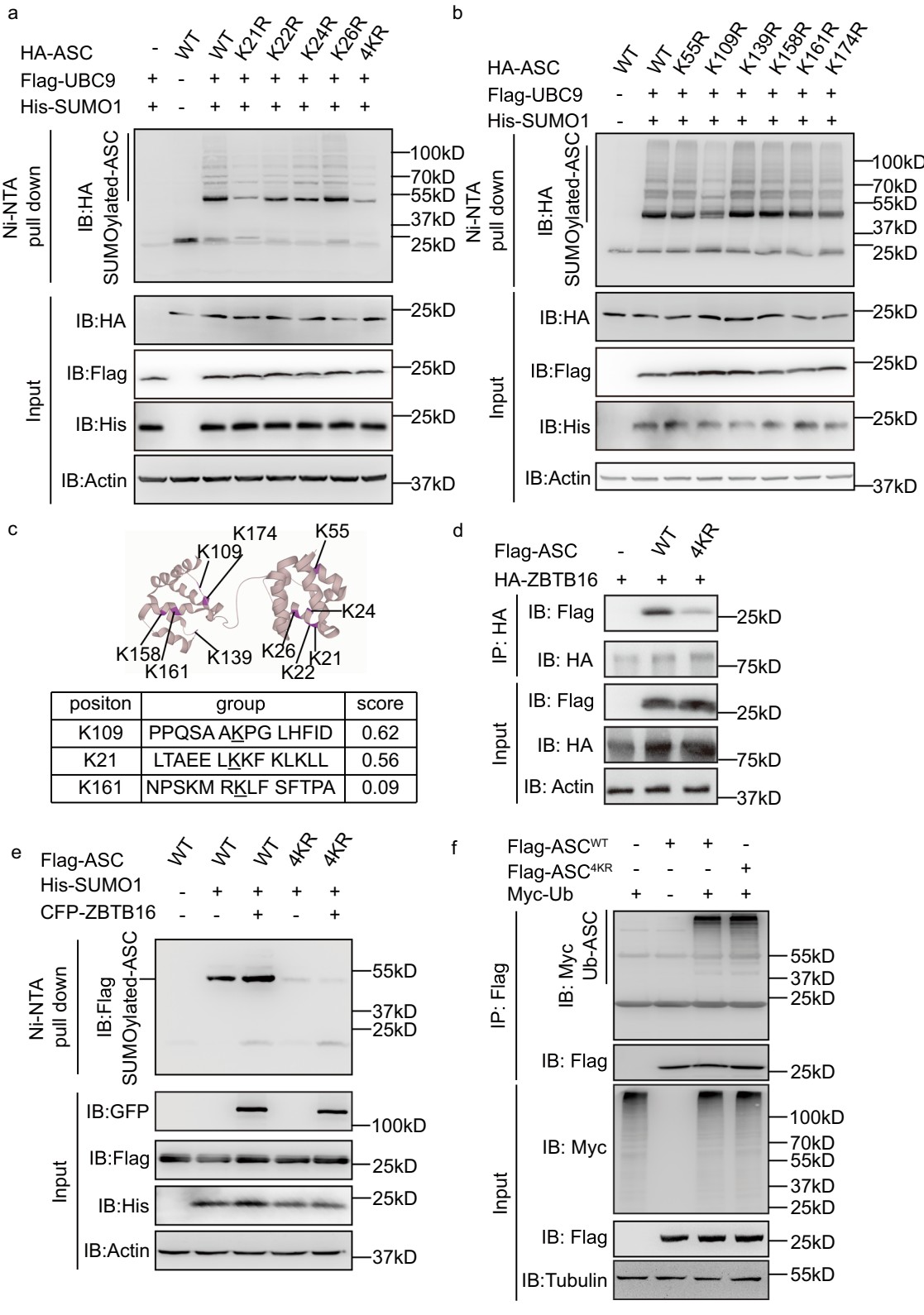

As the lysine residues at positions 21 and 22 of ASC have been identified as being ubiquitinated and because it has been speculated that SUMOylation may function to stabilise proteins by counteracting ubiquitin-mediated degradation[39,40], we sought to assess competition for these four lysine residues by SUMO1 or ubiquitin (Ub). Towards this, the WT or mutant ASC (4KR) was expressed in HEK-293T cells with ZBTB16, UBC9 and Ub. Immunoblotting of immunoprecipitated WT and 4KR ASC with an anti-Ub antibody detected an equivalent signal

(Fig. 7f). This appears to dismiss the notion of competition for the lysine residues between positions 21 and 26 by the ubiquitination complex and is in keeping with the preceding measures that show no effect of ZBTB16 on the levels of ASC (Figs. 2c, d, 4e, 5e and Supplementary Fig. 2a, b, 4, 5b, c with 7g).

We next examined the effect of SUMOylation on ASC oligomerization. This was initially measured with differently tagged ASC constructs in HEK-293T cells. The indirect capture of a Flag-tagged ASC by

**Fig. 7 | The lysine residue at position 21 and 109 on ASC is SUMOylated.** Measures of ASC SUMOylation in HEK-293T cells expressing UBC9, His-SUMO1 and the WT or mutant ASC constructs, with arginine replacement of the lysine residues (**a**) at positions 21 to 26 as separate (K21R, K22R, K24R and K26R) and combined (4KR) mutations or (**b**) positions 55 to 174 of HA-ASC. **c** A representation of the ASC structure (PDB: 2KN6) as a ribbon diagram showing the location of the lysine residues mutated in the study and (below) a table showing the results of a computational (SUMOplot) prediction of SUMOylated residues (underlined) on ASC that are calculated to have a high probability by scoring the match to a consensus sequence that binds UBC9 with substitution of counterpart residues. **d** Arginine replacement of the lysine residues between positions 21–26 on ASC is shown to affect its association with ZBTB16, as assessed in HEK-293T cells expressing HA-ZBTB16 and either the WT or the 4KR mutant Flag-ASC, followed by IP with an anti-HA antibody then IB with an anti-Flag antibody. **e** ZBTB16-dependent ASC SUMOylation is shown to depend on the lysine residues between positions 21–26, as shown by expressing UBC9, SUMO1 and ZBTB16 in HEK-293T cells with either the WT or mutant ASC (4KR), followed by precipitation of His-tagged SUMO1 with Ni-NTA resin, then IB for ASC with an anti-Flag antibody. The expression levels of all constructs are measured by IB with the indicated antibodies. **f** Measures of the ubiquitination of the lysine residues between positions 21–26 of ASC in HEK-293T cells expressing Myc-tagged ubiquitin (Myc-Ub) with either Flag-tagged WT or mutant (4KR) ASC, followed by IP with an anti-Flag antibody and IB with an anti-Myc antibody.

immune-enrichment of an HA-ASC construct was increased by co-expressing SUMO1 with UBC9, as detected by immunoblot of immunoprecipitates (Fig. 8a). Involvement of the lysine residues at positions 21 and 109 in the association between separate ASC proteins was confirmed in the same manner using arginine replacement mutants (Fig. 8b). As an alternative measure, the multimerization of ASC was visualised using constructs tagged with either half of the split-Venus fluorophore. In this instance, a positive effect of SUMOylation on ASC oligomerization was demonstrated by showing that an inhibitor of SUMOylation (2-D08) reduced Venus fluorescence in the cytosol of cells (Fig. 8c). The lysine residues at positions 21 and 109 were also tested for their effect on ASC multimerization upon immune stimulation by expressing the WT or lysine mutant constructs in *Asc*[-/-] BMDMs. Immunofluorescent detection of ASC with an anti-ASC antibody shows the two lysines are important for the formation of ASC specks upon stimulation with LPS and nigericin (Fig. 8d–g and Supplemental Fig. 7f). Immunofluorescent detection of ASC expression in BMDMs verifies that mutation of the lysine residues around position 21 didn't cause retention of ASC in the nucleus, as previously identified for PML-dependent control of ASC (Fig. 8h and Supplemental Fig. 7g)[7].

These data identify that ZBTB16 promotes inflammasome assembly by controlling the modification of the adaptor protein ASC with SUMO1 on lysine residues at positions 21 and/or 109.

## Discussion

We reveal an unexpected role for ZBTB16 in regulating ASC function. As the linchpin of the multiprotein inflammasome, this regulation of ASC is required for full inflammasome activity. This is underscored by our demonstration that targeting ZBTB16 limits pathogenesis in a mouse model of Muckle-Wells syndrome, which is driven by a hyperactive mutant NLRP3. We show that this control extends to acute inflammatory triggers and identify that this is a cell-intrinsic mechanism of action, thereby ensuring the activity is not due to a general immune impairment[41]. Experiments with targeted ablation of *Zbtb16* and biochemical dissection of the response pathway identify that ZBTB16 promotes ASC SUMOylation. Although several reports have identified the importance of ubiquitination and phosphorylation of inflammasome components, including ASC, there is still little known about the regulation by SUMOylation[9–12,42].

SUMOylation proceeds by an enzyme cascade in which SUMO is sequentially processed by the E1 SAE1/SAE2, then E2 UBC9 before ligation to protein substrates with support from E3 ligases. We have not categorically established E3 ligase activity and ZBTB16 lacks the Homologous-to-E6AP-Carboxyl-Terminus (HECT), UFD2-homology (U-box) or Really-Interesting-New-Gene (RING) domains that are associated with E3 activity. However, other BTB-domain-containing proteins, for instance, the Kelch-like ECH-associated protein 1 that forms the Cullin 3 E3 ubiquitin ligase complex, function as substrate adaptor proteins. Our mapping experiments detected that ZBTB16 interacted with ASC, UBC9 and/or SUMO1 via its BTB/POZ and RD2 domains. Accordingly, ZBTB16 may control SUMOylation by modifying noncovalent interactions within an as-yet unrecognised E3 enzyme

complex. Computational (Joined Advanced SUMOylation Site and SIM analyser (JASSA v4) and GSP-SUMO 2.0) analysis predicts a SUMO-interaction motif (ILEIE at residue positions 108-112) on the BTB/POZ domain of ZBTB16 that accords with this activity. Otherwise, ZBTB16 may function more generally to regulate the sequestration of protein partners in order to advance this posttranslational modification.

We identify the lysine residues at positions 21 and 109 on ASC control its SUMOylation. The lysine residue number 21 is part of the interface between ASC oligomerized monomers[38]. Accordingly, it is envisaged that conjugation of SUMO1 at this location would obstruct the formation of the ASC filament unless this was subsequently removed by a SUMO1 protease. Interestingly, it has been shown by others that disruption of protein binding at the PYD of ASC promotes inflammasome activity by increasing the recruitment of pro-Caspase-1 via the proteins' mutual CARDs[43]. Conceivably, SUMOylation of ASC's PYD might function in this way to promote inflammasome assembly.

Notably, ZBTB16 with UBC9 and the best characterised nuclear SUMO E3 ligase PML are all SUMO substrates[7,31,32]. SUMOylation appears to facilitate the formation of membrane-less organelles by phase separation which promotes protein interactions. Phase separation appears as a particular function of PML nuclear bodies in which ZBTB16 localises. Other constituents of the inflammasome such as NLRP3 and cytoskeletal proteins required to assemble inflammasomes in the cytosol are also SUMO1 substrates[30,44,45]. Moreover, cytoskeletal motor proteins such as dynamins, which control inflammasome activity, have been shown to bind SUMO1 and so may act to concentrate SUMOylated proteins[46]. Therefore SUMOylation may promote the colocation of inflammasome constituents.

ASC oligomerization is also regulated by ubiquitination and phosphorylation and two lysine residues that control the association with ZBTB16 have been identified as ubiquitination sites (at positions 21 and 22 or 22 and 23 in human or mouse ASC, respectively[39,40]). However, our experiments did not detect the interplay between ubiquitin and SUMO1. A potential connection to phosphorylation might be envisioned from our previous report that ZBTB16 regulates the NF-κB response, as the IκB kinase and Inhibitor of NF-κB kinase controls ASC[6]. However, no autoregulatory effect on the levels of these kinases was identified in *Zbtb16*[-/-] cells that might alter this phosphorylation activity[22]. Additionally, the residues identified (at positions; 46, 58, 60, 137 and 146 or 16, 144, 187 and 193 in human or mouse ASC, respectively[39,40,47,48]) differ from the residues we identify as controlling ASC SUMOylation and are not contiguous in the three-dimensional structure of ASC.

The SUMOylation of proteins alters their activity, stability and cellular distribution[49–51]. SUMO has been implicated in the development of Amyotrophic Lateral Sclerosis and Huntington's and Parkinson's diseases[52,53], in which the accumulation of fibrillar proteins is a proposed aetiology. There are parallels between amyloidosis and the assembly of immune cell signalling complexes. We speculate that the modification of ASC that we have identified may function as has been identified for the Mitochondrial antiviral signalling (MAVS) protein. In this instance, viral infection induces the formation of a prion-like

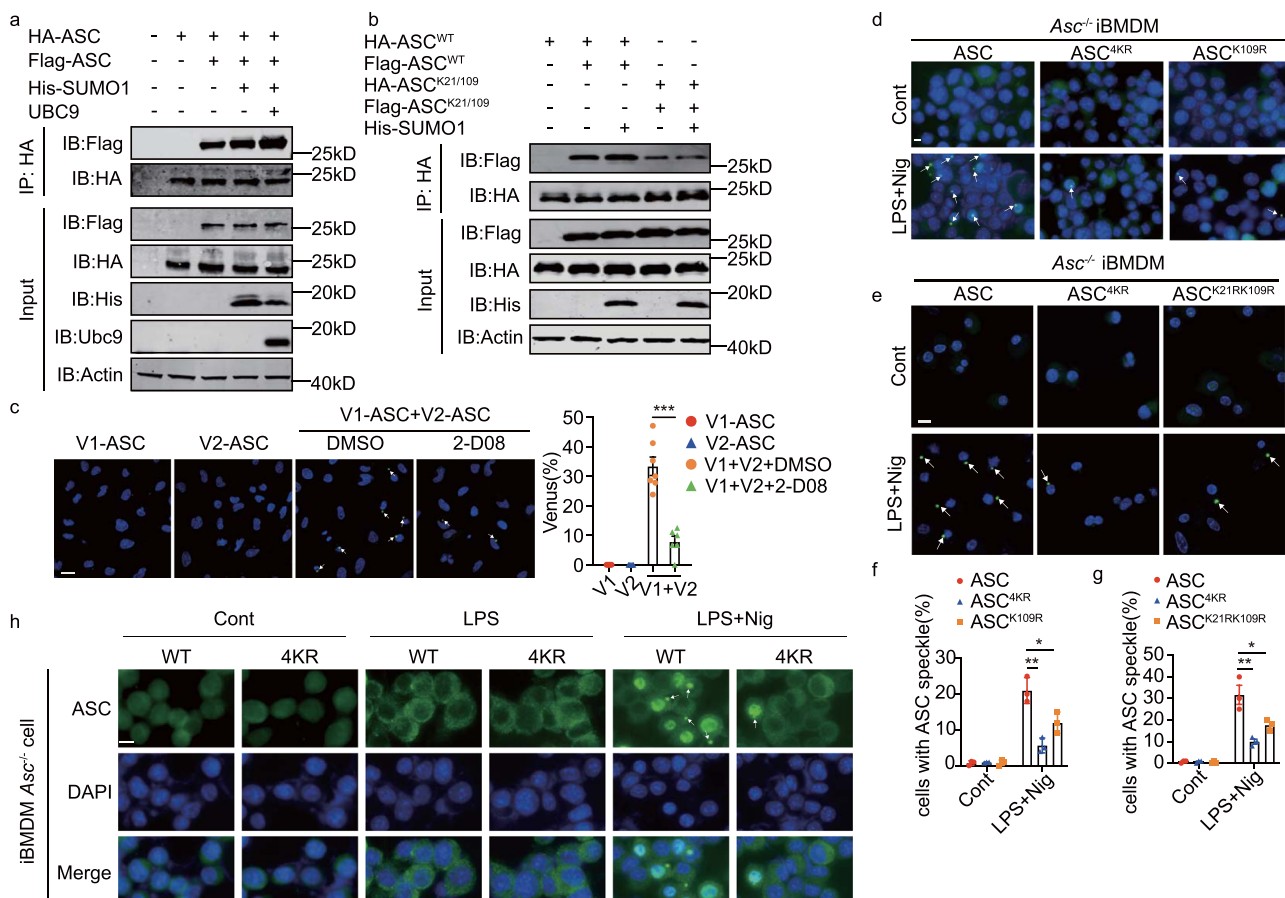

**Fig. 8 | ASC SUMOylation and oligomerization are controlled by lysine residues at positions 21 and 109. a** Detection of the effect of SUMOylation for ASC multimerization in HEK-293T cells transfected with SUMO1, UBC9 and ASC differently tagged with Flag and HA, then IP with an anti-HA antibody followed by IB of SDS-PAGE gel separated immune-enriched proteins with an anti-Flag antibody. **b** Measures of the effect of arginine replacement of the lysine residues at positions 21 and 109 for ASC multimerization, as previously performed. **c** An assessment of the effect of SUMOylation on the oligomerization of ASC by measures of the reconstitution of a full-length fluorophore from each half of a split-Venus (V1 and V2) separately tagged ASC, either without (DMSO) or with a pharmacological inhibitor of SUMOylation (2-D08). Micrographs visualise ASC oligomers in the cytosol as Venus fluorescence (green) and DAPI-stained nuclei (blue) (scale bars = 10 μm) The relative proportion of cells with Venus fluorescence is quantified in the graphs (right) from randomly selected fields ($n = 5$ for V1 and V2, $n = 7$ for V1 + V2 with DMSO and $n = 6$ for V1 + V2 with 2-D08). Data are presented as mean values ± SD. **d–g** Measures of the effect of the lysine residues between 21 and 26 and at position 109 on ASC oligomerization by expressing the indicated WT or arginine replacement mutant constructs (4KR, K109R and K21R + K109R) in immortalised $Asc^{-/-}$ BMDM treated with LPS and nigericin followed by detection of the formation of ASC specks with an anti-ASC antibody. Representative micrographs show the immunofluorescence pattern of the oligomerized ASC (green) in cells with DAPI-stained nuclei (blue) (scale bars = 10 μm). The effect of the different lysine residues on the formation of ASC specks is quantitated in at least 100 cells in 10 randomly selected fields. Data are presented as mean values ± SEM. **h** Micrographs showing the cellular distribution of exogenously expressed WT and the 4KR mutant ASC constructs expressed in immortalised $Asc^{-/-}$ BMDM at rest or upon priming with LPS followed by stimulation with nigericin (LPS+Nig) by immunofluorescent detection with an anti-ASC antibody (green). Cell nuclei were stained with DAPI (blue) (scale bars=10μm). Statistical differences (***$p < 0.001$) were determined by a two-tailed Student's $t$-test ($n = 3$). Source data are provided as a Source Data file.

aggregate of MAVS that is also controlled by its modification with SUMO[54]. Rather than directly promoting MAVS aggregation, it is proposed that SUMOylation promotes the recruitment of signalling components. As SUMO1 appears dispensable for inflammasome assembly, we speculate that ASC SUMOylation isn't directly recruiting protein partners but is partitioning ASC within the cytosol to promote the assembly of inflammasomes (Fig. 9).

These findings advance our understanding of inflammasome activity, recognising that this immune response occurs in concordance with a ZBTB16-dependent SUMOylation of ASC. A growing recognition of the impact of SUMO on the immune response has led to the proposition that inhibitors of the E1 or E2 enzymes or, alternatively, SUMO proteases can constitute immune therapies. However, the quantity and diversity of SUMO substrates suggest it will be difficult to achieve a coherent outcome with such an indiscriminate approach. Greater specificity could be achieved by targeting enzymes with more restricted activity. In this way, autoinflammatory disorders might be suppressed by inhibiting ZBTB16 or, on the other hand, anti-tumour immunity might be advanced by pharmacologically promoting ZBTB16 activity[55–57]. Accordingly, our findings identify a potential drug-targeting strategy for immune therapy by controlling inflammasome activity.

## Methods

### Mice

$Zbtb16^{-/-}$ mice were reported by our laboratory previously[21,22]. $Zbtb16$ conditional knockout mice were generated by BIOCYTOGEN (Beijing, China). Using CRISPR/Cas9-mediated genome editing on a C57/BL/6 J background, as described in the supplementary data (Supplementary Fig. 1). Lysozyme M-Cre knock-in mice (CreL) were obtained from the Jackson Laboratory. $Nlrp3^{R258W}$ and $Asc^{-/-}$ mice were reported previously[58,59]. $Nlrp3^{R258W}$ mice were maintained as heterozygotes by backcrossing with wild-type (WT; $^{+/+}$) control mice from the Jackson Laboratory. Female and male mice aged between eight to ten weeks were used. All mice in the C57BL/6 background were maintained in

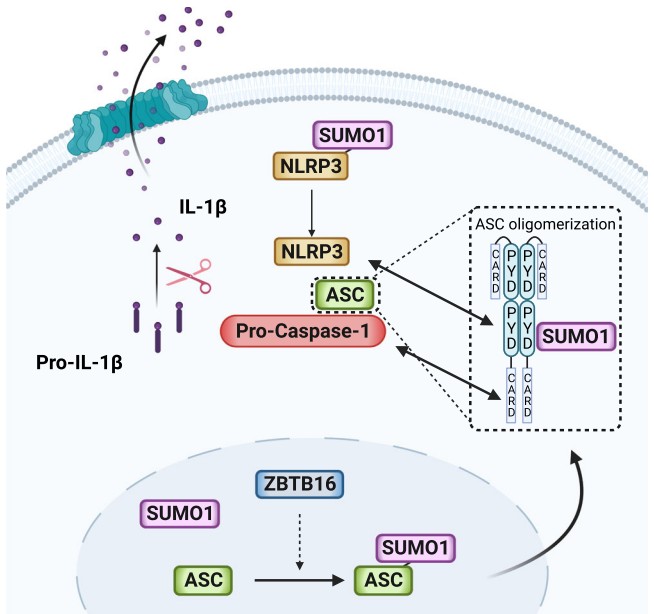

**Fig. 9 | Inflammasome activity is promoted by ZBTB16-dependent SUMOyla-tion of ASC.** A synopsis of the immune activity we report, in which inflammasome assembly progresses by a ZBTB16-SUMO1-ASC axis. ZBTB16 interacts with ASC in the nucleus to promote its modification with SUMO1. SUMOylated ASC then translocates to the cytosol where it may be deSUMOylated by SUMO proteases to re-enter the nucleus or, upon immune stimulus, assembles the inflammasome. We speculate that ASC SUMOylation alters its homologous and heterologous protein interactions and/or partitions ASC in the cytosol with other inflammasome con-stituents, some of which such as NLRP3 are also modified with SUMO1, to promote the inflammatory response.

specific pathogen-free conditions at the Animal Center of Hangzhou Normal University (Zhejiang, China). Mice were maintained at a temperature of 21 °C and humidity of 50-70% and under a constant 12 h light/12 h dark cycle with continuous access to food and water. All experiments were approved by the IACUC at Hangzhou Normal University and Ruijin Hospital, Shanghai Jiao Tong University School of Medicine.

## Cell culture
Primary bone marrow-derived macrophages (BMDMs) were derived from bone marrows of 6−10 weeks old C57BL/6 WT and *Zbtb16*[-/-] mice were prepared by methods described previously[21]. Briefly, bone marrow cells were prepared from mice and cultured in RPMI-1640 medium supplemented with 10% foetal bovine serum (FBS), 30% L929-conditioned medium and antibiotics (100 units/mL penicillin and 100 μg/mL streptomycin). After 4 days of culture, nonadherent cells were removed and a fresh culture medium was added. On day 6 cells were seeded into 6 well plates at $1.5 \times 10^6$ cells/well for experimentation on day 7. Immortalised WT and *Asc*[-/-] BMDMs (gifted by Dr Eicke Latz, Institute of Innate Immunity, University of Bonn) were cultured in DMEM supplemented with 10% FCS and 10 μg/mL Ciprobay-500. Peritoneal macrophages were collected by lavage with 10 mL RPMI-1640 medium. The cells were subsequently plated in 96-well plates and cultured overnight for further study. HEK293T, HeLa and THP-1 cells were purchased from ATCC and were grown in DMEM (HEK293T and HeLa) or RPMI-1640 (THP-1) with 10% FBS with antibiotics. All cells were cultured in a 5% $CO_2$ incubator at 37 °C.

## Transfection and lentiviral reconstitution of immortalised macrophages
For exogenous gene expression in HEK293T and HeLa, cells were transfected with EZ-Trans cell Transfection Reagent (Life-iLab)

according to the manufacturer's instructions. For gene expression in macrophages, cells were transduced with recombinant lentivirus pre-pared by transiently transfecting lentiviral vectors (pCDH-puromycin, Δ8.91 and VSV-G) expressing WT or mutants of ASC into HEK293T cells for 48 h[60]. Transduced macrophages were selected with 5 μg/mL of puromycin for 7 days and protein expression was validated by immunoblotting.

## Generation of CRISPR/Cas9 inducible knockout cell line
An inducible CRISPR (clustered regulatory interspaced short palin-drome repeats) /Cas9 (CRISPR-associated 9) lentiviral system was used to construct THP-1 *ZBTB16* mutant lines. THP-1 cells were infected by lentiviral particles containing the pFUCas9-mCherry vector and sgRNAs targeting exon 2 of *ZBTB16*, cloned into the doxycycline (DOX)-inducible vector pFgh1tUTG-GFP. THP-1 cells expressing Cas9 and sgRNA were isolated as GFP and mCherry positive cells by flow cytometry (FACS Aria II, BD Biosciences)[61]. Gene knockout was induced by the culture of transduced cells with 1 μg/mL DOX (Sigma) for 5 days and confirmed by measuring *ZBTB16* mRNA levels by RT-PCR (sgRNA and RT-PCR primer sequences are shown in Supplementary Table S1).

## Inflammasome activation
BMDMs or THP-1 cells treated with 20 ng/mL PMA were plated in 6 well plates and 24 h later primed with 100 ng/mL LPS (Sigma) for 4 h before being treated to activate the inflammasome. Activation was by stimu-lated with Nigericin (20 μM) for 30 min; ATP (5 mM) for 30 min; silica (120 μg/mL) for 6 h, or MSU (200 μg/mL) for 6 h (InvivoGen). For *Clostridioides difficile (C. difficile)*-related inflammasome activation, cells were stimulated with freshly cultured *C. difficile* 630 (ATCC BAA1382) at a multiplicity of infection (MOI) of 100 for 6 h or primed with LPS (1000 ng/mL) for 2 h followed by 200 ng/mL of the microbe toxin B (Abcam) for 4 h.

## Reagents
Details of Flag-ZBTB16, CFP-ZBTB16, His-SUMO1/2/3, HA-UBC9 and SENP1 have been reported previously[22,62]. Plasmid pLV-mTurquoise2-IL-1β-mNeonGreen was a gift from Dr Hao Wu (Harvard Medical School, Addgene number 166783). HA-ASC and Flag-ASC plasmids were gifted by Dr Eicke Latz (Institute of Innate Immunity, University of Bonn). Mutants of ASC were generated by PCR-mediated, site-directed mutagenesis. PCR primers for gene cloning and site-directed muta-genesis are listed in Supplementary Table S2.

## Measurement of cytokines
Supernatants from cell culture, peritoneal lavage fluid and serum were collected and measured by ELISA as per the manufacturer's instruc-tions: Mouse IL-1β (eBiosciences); Mouse IL-18 and TNFα and human IL-1β (R & D Systems).

## Lactate dehydrogenase assay
The release of lactate dehydrogenase (LDH) from cells was measured by LDH Cytotoxicity Assay according to the manufacturer's instruc-tions (Beyotime Biotechnology). Measurements were performed in duplicate.

## ASC oligomerization assay
BMDM cells were permeabilized with TBS buffer (50 mM Tris-HCl and 150 mM NaCl) with 0.5% Triton X-100 with protease inhibitors (Thermo Fisher Scientific) and lysed by passing through a 21-gauge needle, then centrifuged for 10 min at 4 °C. The pellets and the supernatants were used as Triton-insoluble and soluble fractions. The Triton-insoluble fractions were further resuspended in TBS buffer, then cross-linked by treatment with 2 mM disuccinimidyl suberate (DSS) (Thermo Fisher Scientific) at room temperature for 30 min. The lysates were analysed

by 12.5% SDS-polyacrylamide gel electrophoresis (PAGE), followed by western blotting assays with rabbit anti-ASC antibody.

## Immunoprecipitation and western blotting analysis

Cells were lysed in triple detergent lysis buffer (Thermo Fisher Scientific) and then incubated with antibody-conjugated agarose beads at 4 °C overnight, or with unconjugated antibody for 2 h followed by overnight incubation with protein A/G-agarose beads (Beyotime Biotechnology). After three washes with lysis buffer, proteins were heat-denatured in 2x SDS sample buffer, then separated by 8–15% polyacrylamide by SDS-PAGE and transferred to Immobilon-FL Membrane (Millipore). Membranes were probed with primary antibodies as listed in Supplementary Table S3, then incubated with LICOR IRDye secondary antibodies diluted 1:20,000. Immune complexes were visualised and quantified with the Odyssey Imaging System (LI-COR, USA) or incubated with a peroxidase-conjugated secondary antibody and detected by enhanced chemiluminescence (Beyotime Biotechnology). Full blots of images cropped for presentation are presented in Supplementary Fig. 8.

## SUMOylation analysis

ASC SUMOylation was analysed in HEK-293T cells by enriching His-SUMO using nickel-nitrilotriacetic acid (Ni-NTA) coupled agarose beads or enriching HA-ASC with HA Nanoab Mag Beads as previously described[33]. Endogenous SUMOylated ASC were enriched by immunoprecipitations following the published protocol[63,64] with minor changes. Briefly, $5 \times 10^7$ of cells were lysed in 1 mL of lysis buffer (20 mM Phosphate buffer pH 7.4, 150 mM NaCl, 1% Triton, 0.5% Na-deoxycholate, 1% SDS, 5 mM EDTA, 5 mM EGTA, 1 µg/mL aprotinin, 1 µg/mL leupeptin, 1 µg/mL Pepstatin A, 0.1 mM Pefabloc, 20 mM N-ethylmaleimide (NEM), protease and phosphatase inhibitors (Sigma-Aldrich)). The lysate was sonicated to reduce its viscosity, then diluted 1:10 with RIPA buffer without SDS and incubated with antibody-coupled beads overnight at 4 °C. Beads were washed three times with high-salt buffer and then boiled for SDS-PAGE immunoblotting analysis.

## Quantitative real-time PCR

Total RNA was extracted using the RNeasy Plus Mini Kit (QIAGEN) and treated with a TURBO DNA-free Kit (Ambion). cDNA was synthesised from RNA using the PrimeScript RT Reagent Kit (TaKaRa). Quantitative PCR (Q-PCR) was performed with SYBR Green (Applied Biosystems) using an ABI 7700 Prism real-time PCR instrument. The expression of mRNA was normalised to the expression of either 18 S ribosomal RNA or GAPDH as the change in cycling threshold (ΔCT) method and calculated based on 2-ΔCT. Results are articulated as relative gene expression for triplicates of each target. PCR primers used are listed in Supplementary Table S4.

## Immunofluorescence

Cells were seeded at a concentration of $2 \times 10^5$ cells/well onto round coverslips in 24-well plates. After treatment, cells were fixed in 4% paraformaldehyde for 30 min and blocked and permeabilized in 500 µl blocking buffer (PBS, 10% FBS, 0.5% Triton X-100) for 1 h at room temperature, before incubation with the primary antibody in blocking buffer overnight at 4 °C. After washing with PBS three times, cells were treated with a dilution of 1:200 of the Alexa-488 or Alexa-594 conjugated secondary antibody for 1 h. Nuclei were co-stained with 4′,6-diamidino-2-phenylindole (DAPI, from Cell Signalling Technology). The images were captured with a Zeiss LSM710 confocal fluorescence microscope with objective Plan-Apochromat 63×/1.40 oil DIC M27 objective. Z-stack digital images were acquired with ZEN imaging software (Carl Zeiss, GmbH). ASC specks were counted in five random areas of each image in triplicate experiments, ensuring a minimum of 100 cells from each treatment condition.

## Colocalization measurement

Single-plane images of wild-type and *Zbtb16* knockout cells stained with DAPI, anti-ASC and anti-SUMO antibodies were captured by confocal microscopy. The co-localisation score was calculated using the GcoPS tool in ICY[65] using a threshold with constant intensity cut off on the DAPI/nuclear channel as a mask for calculating colocalization in the nucleus. The Pearson's coefficient (colocalization) was calculated using the nuclear or H-K-means clustering (intensity cut-off: 100, clusters: 9, min/max size 100/1000) of either the DAPI stain or SUMO antibody (intensity cut-off: 80, clusters 9, min/max size 100/300). The data is graphed as the mean Pearson's coefficient per individual image series with SEM with each image containing up to 10 cells.

## Image processing and quantification analysis

The Z-sections of images acquired by confocal microscopy were imported into IMARIS software for 3D surface rendering and quantitative analysis. Surfaces were generated by selecting the source channel and setting the smooth surface detail to 0.2 µm. Background subtraction was set to 0.6 µm. The threshold for surfaces was adjusted to ensure complete coverage of all voxels and the surface area and volumes of the generated surfaces for both channels were quantified with IMARIS. The overlap between the 3D surface-reconstructed images (for ASC and SUMO1 or ZBTBT16 and ASC) was processed using the IMARIS surface-surface overlap module. This module tracked the overlapping surfaces and calculated the areas of surface colocalization. Data was exported to Excel for further analysis.

## Bimolecular fluorescence complementation assay

Gene open reading frames (ORFs) were cloned into the AccIII-XbaI or NotI-ClaI restriction endonuclease sites of pcDNA3-V1/V2 to produce either amino- or carboxyl-terminal split-Venus-tagged proteins, respectively. The ZBTB16 constructs were established previously[24]. HEK-293 cells were seeded into a 24-well culture plate and transfected with 50 ng/well of each construct using lipofectamine 2000 (Thermo Fisher Scientific). Total Venus fluorescence was captured 30 hours after transfection of cells using a Zeiss MBQ 52 AC burner with a 4x lens on an Axiovert 40 CFL Trinocular Inverted fluorescence microscope and a ProgRes camera and software (Jenoptik). The level of fluorescence was normalised to a nonfluorescent control constituted by coexpressing split-tagged ZBTB16 and the Eukaryotic initiation factor 2α and/or SUMO1 tagged with each half of the split fluorophore. The level of fluorescence was averaged from three images per experiment. The data was analysed using the ImageJ software (NIH and LOCI). Micrographs were captured from cells seeded and transfected on sterile coverslips (Menzel-Glaser, 12 mm diameter, #1.5, Thermo Fisher Scientific) within a 24-well culture plate. After 30 h cell nuclei were stained with Hoechst (Invitrogen), washed with PBS then fixed with 10% formalin for 20 min, rewashed with PBS and mounted in; 13% Mowiol, 33% glycerol, and 20% sodium azide (pH 8.5), then recorded with an Olympus U-RFL-T burner, BX60 microscope using a DP74 camera and the Olympus CellSens software.

## Proximity ligation assay

The interactions of endogenous ASC, ZBTB16, SUMO1 and NLRP3 were assessed in macrophage cells by proximity ligation assay (PLA, Duolink) according to the manufacturer's instructions (Sigma-Aldrich). BMDMs were fixed with ⁻20 °C methyl alcohol and then permeabilised with 0.3% Triton X-100 before the addition of the specified antibody pairs mouse anti-ASC (Santa Cruz) with rabbit anti-ZBTB16 (Bioss) or rabbit anti-SUMO1 (Cell Signalling Technology) or, alternatively, rabbit anti-ASC (AdipoGen) with mouse anti-NLRP3 (AdipoGen). The images were captured by Zeiss LSM 880 confocal fluorescence microscope at 63x magnification with Airyscan image processing and analyses with ImageJ software.

## Peritonitis model

Peritonitis was induced by intraperitoneal (IP) injection with 3 mg MSU crystals (InvivoGen) in 300 µl PBS or PBS alone as a control. After 6 h, the mice were sacrificed and cells from the abdominal cavity were collected with 500 µl PBS lavage. Cells were collected by centrifuging at 300 g for 5 min and subjected to flow cytometry analysis. The supernatant was subjected to ELISA assay. The tissues of the peritoneum were subjected to histochemical staining.

## DNCB-induced skin inflammation

A 2 cm² area of skin was shaved in the middle of the back of 6 to 10-week-old mice and a solution of 5% 1-chloro-2,4-dinitrobenzene (DNCB, Sigma) dissolved in acetone/olive oil was applied. Five days after this sensitisation treatment the mice were rechallenged by applying 0.5% DNCB on each side of both ears. The skin response was assessed 24 h after rechallenging by measuring the interfollicular thickness of the epidermis, hematoxylin and eosin (H&E) staining and histology, as well as immunostaining with specific antibodies.

## Histochemical staining and immunohistochemistry

Tissue samples obtained from the peritoneum of mice were fixed in 4% phosphate-buffered formaldehyde and then paraffin processed and embedded. Cut sections were stained with Masson's trichrome stain. Tissues from the skin inflammation mouse model were formalin-fixed and paraffin-embedded, then sectioned for H&E staining with image capture by Leica DM IL microscope. Alternatively, skin sections were deparaffinized and antigen retrieval was performed by heating in citrate buffer (pH 6.0). Endogenous peroxidases were inhibited by incubation with 0.9% hydrogen peroxide and blocking was performed using 3% BSA. Sections were incubated with mouse anti-Ly6G antibody (Bio X cell), or Anti-S100A9 (Cell Signalling Technology). Antibody binding was detected with an HRP-labelled secondary antibody (Santa Cruz Biotechnology Inc.) and visualised by the DAB+ Substrate Chromogen System (Dako Omnis). Samples were counterstained with hematoxylin, incubated in ethanol and xylol solutions of ascending concentrations then mounted. Images were captured with a Nikon DS-F2 microscope or stained slides were scanned at 20 x magnification using the Aperio ScanScope XT imaging system (Aperio, Vista) and analysed using the ImageScope software.

## Myeloperoxidase activity assay

Myeloperoxidase (MPO) activity was evaluated as an index of neutrophil accumulation in the tissues by using the MPO Activity Assay Kit (BioVision) following the manufacturer's protocol.

## Flow cytometry analysis

Single-cell suspensions from peritoneal lavage were stained with APC-CD45, PerCP-Cy5.5-CD11b and PE-Ly6G (BioLegend). Data were acquired on the FACSCanto II (BD Biosciences) and analysed by FlowJo Software.

## Statistical analysis

All experiments were independently performed at least three times in triplicate. Statistical analysis was performed with GraphPad Prism 8.0 software. The statistical difference between the two groups was performed by the unpaired two-tailed $t$-test or analysis of variance. $P$ values of less than 0.05 were considered significant and different levels of significance were expressed as follows: *$p < 0.05$; **$p < 0.01$; ***$p < 0.001$.

## Reporting summary

Further information on research design is available in the Nature Portfolio Reporting Summary linked to this article.

## Data availability

The main data supporting the findings of this study are available within the article and its Supplementary Information or from the corresponding author on request. Source data are provided with this paper.

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

## Acknowledgements

We gratefully acknowledge Eicke Latz (Institute of Innate Immunity, University Hospital Bonn, University of Bonn) for generously providing *Asc*[-/-] BMDM cells, Dr Jing Yi (Shanghai Jiao Tong University School of Medicine) for the gifts of His-SUMOs and SENP1 plasmids, Dr Ashley Mansell (Hudson Institute of Medical Research, Monash University) for an ASC reporter cell line, and Dr Warren Strober (NIH) for sharing the *Nlrp3*[R258W] mouse line. We also thank members of the Hui Xiao laboratory for providing technical expertise and reagents. We thank Chenzhi Guo, Ying Huang and all members of the Core Facility of Basic Medical Sciences of Shanghai Jiao Tong University for their technical support. This work was supported by grants from the National Natural Science Foundation of China (82071811, 82301978, 81871274, 82372306, 81902117, 82172324, 81971993 and 81871715) and the National Health and Medical Research Council of Australia (1143839). Figure 9 was created with BioRender.com.

## Author contributions

D.D., X.F. and Y.D. performed most of the in vitro and in vivo experiments; A.S., H.Y., X.Li, X. Yang., J.M., S.H., Z.M. and J.Z. performed in vitro experiments; X.Yang. and Z.M. performed in vivo experiments; Y.P., D.W.C., L.S., Y.L., A.T.I., X.Yuan., X.Liu., P.N., Y.H. and G.M. contributed reagents and helped to design the experiments; Y.P. helped to edit the manuscript A.S. and D.X. wrote the manuscript. DX conceived the study, oversaw the experiments, analysed the data and provided overall direction.

## Competing interests

The authors declare no competing interests.
