## [Peer Review File · Nature Communications]

Inflammasome activity is controlled by ZBTB16-dependent SUMOylation of ASCREVIEWERS' COMMENTS:

Reviewer #1 (Remarks to the Author):

This manuscript is focused on inflammasome activity and how it is controlled by the adaptor ASC, which in turn is controlled by SUMOylation via ZBTB16. However, by the end of the paper, it becomes evident that the authors are suggesting that ZBTB16 mediates the interaction of ASC with SUMO1 which then SUMOylates ASC.

Major concerns:

- The manuscript contains many sentences that are difficult to understand because of the grammar mistakes or the misuse of verbs.
- The way the paper is structured, gave the impression that ZBTB16 directly mediates the SUMOylation of ASC.
- However, in the last few figures, it seems that the authors propose that SUMOylation of ASC requires SUMO1?
- For example, the sentence “discovery that ZBTB16 interacts with ASC in the nucleus to control its activity by post-translational modification” is confusing. Did they mean that the post translational modification is directly mediated by ZBTB16? Then, later in the paper, “they identify that ZBTB16 interacts with ASC in PML bodies to facilitate its modification with one of the three SUMO paralogs, SUMO1”
- They mention in the M&M section immortalized WT and ASC^{-/-} BMDMs, however their source and their use were not clarified.
- The conclusion that ZBTB16 interacts with ASC are based on experiments that lack essential controls. For example, using cross linkers are known to generate false positive interactions. The authors did not use a negative control to rule out this possibility.
- Using over-expressed proteins also promotes false interactions, yet this concern was not remedied.
- The cross linker was used to show ASC oligomerization, however, ZBTB16 was not detected, hence, how do they suggest later that they interact?
- The authors do not put any molecular weight signs on any western, which makes it hard to interpret.
- All the confocal images are of very poor quality, the colocalization of red and green is not accurate due to overexpression of the two colors. In addition, to confirm colocalization, 3D

and Z stack analyses are needed.

- The authors use the term ASC activity several times, however, it is not clear what is meant by its activity?

- Using the production of IL-1B in the supernatant as a measure for inflammasome activity is incorrect, since it is well known now that IL-1B release into the supernatant relies on many factors that may be unrelated to the activation of the inflammasome. Accordingly, in some of their westerns, when IL-1B release is reduced, there is more accumulation inside the cell. Also, it usually accompanies LDH release which can be due to other caspases.

- They used the FLICA assay to measure caspase-1 activity, which is known to be non-specific.

- It is not clear why the reduction in IL-1B production in Nlrp3R258W/Zbtb16-/- means that Zbtb16 acts downstream of Nlrp3.

- Immunoblotting for immunoprecipitated proteins can suggest that 2 molecules are part of a complex, but it does not mean that they directly interact. Negative controls for all IP experiments were not included.

- The authors suggest that ASC, ZBTB16 and SUMO1 interact but they did not present any data that actually demonstrate that SUMO1 SUMOylates ASC.

Other concerns:

-Peritoneum pathogenesis was not scored

-Methods of statistical analysis of every panel should be added to the corresponding legends with the number of Ns

Reviewer #2 (Remarks to the Author):

The manuscript suggests interesting new functions and roles of ZBTB16/PLZF in inflammasome activity. The authors found ZBTB16 is important for promoting NLRP3 inflammasome activity by ZBTB16-deficient mouse experiments. ZBTB16 is known as a transcription factor, but the authors suggest that ZBTB16 is likely to have a different function in the inflammasome activity. The authors have shown that ZBTB16 binds and SUMOylates ASC (apoptosis-associated speck-like protein containing a CARD). The authors conclude that ZBTB16 positively regulates inflammasome activity via SUMOylation of ASC

and the subsequent oligomerization of ASC. The data might contribute to the development of new therapeutic strategies to target inflammasome.

Almost all of the experiments are excellent and the evidence for the conclusion is convincing. The paper warrant publication in this journal after addressing several comments listed below.

Comments:

(1) In the discussion section, the authors suggest that ZBTB16 acts as a SUMO E3 ligase. The suggestion is very interesting, but it is necessary to access the E3 ligase activity of ZBTB16 using recombinant proteins.

(2) In Fig 4a, the blot against ASC is unclear. It is better to re-examine this experiment.

(3) In Fig 5h, the signal of ASC in WT is weak. Is it difficult to detect SUMOylated-ASC?

(4) Recent papers reported that ZBTB16 can be degraded by several drugs like thalidomide derivatives (Matyskiela et al. ACS Chem Biol 2020, DOI: 10.1021/acscchembio.0c00674, Yamanaka et al. EMBO J 2021, DOI: 10.1021/acscchembio.0c00674, Shimizu et al. Commun Biol 2021, DOI: 10.1038/s42003-021-02801-y.). It might be better to discuss the therapeutic approach against pathogenic inflammasome activity by referring to these papers.

Reviewer #3 (Remarks to the Author):

In the manuscript the authors claim that ASC activation and oligomerization is controlled by ZBTB16-mediated SUMOylation. However, the data do not support this conclusion. Thus, although the data demonstrate the SUMOylation of ASC, its relationship with protein oligomerization or its regulation by ZBTB16 is not probed. The lack of molecular weight markers in many experiments as well as the lack of essential controls makes it difficult to interpret the results. There are multiple error in both the text and figures.

Main criticisms:

Figure 4a. Molecular weight markers are essential to evaluate the results. The authors claim that ZBTB16 modulates ASC oligomerization; however, a reduction in the intensity of all the bands detected with the anti-ASC antibody (labelled as monomer, dimer and oligomer) is

observed in *Zbtb16*^{-/-} cells. Moreover, why the total amount of ASC protein (insoluble and soluble fractions) in WT cells treated with LPS and Nig is clearly higher than in LPS and Nig treated *Zbtb16*^{-/-} cells or in untreated cells? In panel d no differences are observed in the levels of ASC between WT and *Zbtb16*^{-/-} cells.

Figure 4b. It is not clear from the pictures if there is less ASC speckles in *Zbtb16*^{-/-} cells or if the levels of the ASC are lower in these cells.

Figure 5a. It is not clear to me that the ASC-ZBTB16 co-localization takes place in the nucleolus as the authors say. What happens in untransfected cells?. Does ZBTB16 overexpression induce ASC translocation? Why the subcellular distribution of ASC in Figure 5a and 5i is totally different from the one shown in Figure 7i?

What happens with the ASC-ZBTB16 or ASC-SUMO co-localization in WT and *ZBTB16*^{-/-} cells after LPS+Nig treatment?

Figure 5b. Why immunoprecipitated HA-ASC protein is detected as a double band?. The experiments shown in figures 5b and 5c lack essential controls (immunoprecipitation using a control antibody)

Figure 5d. Molecular weight markers are essential to interpret the results. In addition, analysis of the protein extracts after pull down using anti-Histidine antibody is also essential to interpret the data. As a result of this experiment the authors conclude that ASC is mainly modified by SUMO1. However, the levels of unmodified Flag-ASC protein detected in the pull-down extracts is lower in those cells transfected with SUMO2 or SUMO3. Therefore, the absence of bands corresponding to SUMO2 or SUMO3 modified ASC protein in the blot could be due to a lower level of protein in those lanes.

Figure 5e. Molecular weight markers are essential to interpret the results. In addition, analysis of the protein extracts after pull down using anti-Histidine antibody is also essential to interpret the data.

Figure 5f. Molecular weight markers are essential to interpret the results. In addition, analysis of the immunoprecipitated proteins using anti-SUMO1 antibody is also essential

evaluate the data.

Figure 5g. Molecular weight markers are essential to evaluate the results. In addition, analysis of both input and Ni-NTA pulldown extracts with anti-Histidine antibody is also essential to interpret the data.

Figure 5h. Molecular weight markers are essential to analyze the results. In addition, analysis of the immunoprecipitated proteins using anti-SUMO1 antibody is also essential to interpret the data. Why is the pattern of the ACS protein detected in the SUMO1 immunoprecipitated extract in this figure totally different from the one shown in figure 5f?

Figure 5i. It seems that the SUMO1 signal in the ZBTB16^{-/-} cells is lower than in the WT cells. The WB analysis to compare the levels of SUMO1 present in both cells included in supplementary information should show both free and conjugated SUMO1 protein. The presence of lower levels of SUMO1 in ZBTB16^{-/-} cells could explain a lower co-localization with ASC.

Does ZBTB16 promote the SUMOylation of ASC in vitro?

Figure 6a. The number of fluorescent dots is different in each panel. Moreover, cells transfected with ASC-ASC plasmids only show one dot. How significant is this data? Western blot analysis on lysates of transfected cells in each BiFC experiment is required to determine if both fusion proteins are expressed equivalently. In addition, analysis of the subcellular localization of each fusion protein using antibodies specific for each fusion protein is also essential to evaluate whether the fusion of the fluorescent fragment alters the subcellular localization of the protein of interest.

Figure 6b. Representative images of cell transfected with the different plasmids is required. It is not clear how the fluorescence was quantified. Western blot analysis on lysates of transfected cells in each BiFC experiment is required to determine if both fusion proteins are expressed equivalently. Do ASC and SUMO1c co-localize?

Figure 7b. Molecular weight markers are required to interpret the results. In addition,

analysis of the protein extracts after pull down using anti-Histidine antibody is also essential to interpret the data. WB of the whole cells extracts with anti-Flag antibody is also required. The authors conclude that mutation of some specific lysine residues reduces ASC SUMOylation. However, the intensity of the bands corresponding to the unmodified mutant proteins is less than that of the unmodified WT protein. Therefore, a reduction in the intensity of the SUMOylated bands could be due to a lower level of protein in the Ni-NTA pull down extracts.

Figure 7c. Molecular weight markers are essential to analyze the results. In addition, analysis of the protein extracts after pull down using anti-Histidine antibody is also essential to interpret the data. WB of the whole cell extracts with anti-Flag antibody is also required. Mutation of K109 seems to reduce SUMOylation. Why haven't the authors made a mutant by combining this lysine with the mutated ones in Figure 7b?

Figure 7d. Molecular weight markers are essential to interpret the results. In addition, analysis of the protein extracts after pull down using anti-Histidine antibody is also essential to interpret the data. Several panels in Figure 5 show also WB analysis of Flag-ASC protein in His-SUMO1 pull-down extracts. In all those panels, bands corresponding to both the unmodified and modified ASC protein can be observed. Why is only one band detected in this blot?. The authors state that the 4KR has lost the SUMOylation. However, SUMOylation of the 4KR mutant is clearly observed.

Figure 7f. Molecular weight markers are essential to evaluate the results. In addition, analysis of the protein extracts after pull down using anti-Histidine antibody is also essential to interpret the data.

Figure 7g. Molecular weight markers are required to interpret the results. Is this IP carried out in denaturing or non-denaturing conditions? If it has been carried out in non-denaturing conditions, the bands detected with anti-ubiquitin antibody could correspond to the ubiquitinated proteins that interact with ASC instead of the ubiquitinated-ASC protein.

Figure 7h. Evaluation of the oligomerization state of the ASC mutants is essential to

understand the involvement of SUMO on the regulation of ASC oligomerization.

Figure 7i. In those cells treated with LPD and Nig, the signal of the 4KR mutant is clearly lower. Is the protein being degraded?

Figure 8. The authors propose that ASC is modified by SUMO in the PML-NBs. However, in the manuscript the authors state that PML and ZBTB16 co-localize inside the nucleolus?

Response to Reviewers' comments:

Reviewer #1

R1: This manuscript is focused on inflammasome activity and how it is controlled by the adaptor ASC, which in turn is controlled by SUMOylation via ZBTB16. However, by the end of the paper, it becomes evident that the authors are suggesting that ZBTB16 mediates the interaction of ASC with SUMO1 which then SUMOylates ASC. [And latter] The way the paper is structured, gave the impression that ZBTB16 directly mediates the SUMOylation of ASC. However, in the last few figures, it seems that the authors propose that SUMOylation of ASC requires SUMO1?

Authors: There appears to be confusion about the process of SUMOylation. SUMO proteins are covalently attached to target proteins. Our data identify that ZBTB16 promotes the modification of ASC with SUMO1 (by the activity of the E1 and E2 enzymes).

R1 Major concerns: The manuscript contains many sentences that are difficult to understand because of the grammar mistakes or the misuse of verbs.

Authors: The manuscript has been reviewed by a professional English service.

R1: The sentence “discovery that ZBTB16 interacts with ASC in the nucleus to control its activity by post-translational modification” is confusing. Did they mean that the post translational modification is directly mediated by ZBTB16? Then, later in the paper, “they identify that ZBTB16 interacts with ASC in PML bodies to facilitate its modification with one of the three SUMO paralogs, SUMO1”.

Authors: The text is not faithfully quoted, and the section enclosed in quotes is not from our manuscript, which adds to the confusion. The (presumed) sentences' now read; “ZBTB16 interacts with ASC in the nucleus to promote its modification with SUMO1” and “This is shown to facilitate the modification of ASC with one of the three SUMO paralogs, SUMO1”. Hopefully, this clearly communicates the idea that ZBTB16 promotes the post-translational modification of ASC with SUMO1.

R1: They mention in the M&M section immortalized WT and ASC^{-/-} BMDMs, however their source and their use were not clarified.

Authors: Eicke Latz from the Institute of Innate Immunity, University Hospital Bonn, University of Bonn is the source of the ASC^{-/-} BMDM. This information had been given in the acknowledgements and is now repeated in the Methods section in the revised manuscript.

R1: The conclusion that ZBTB16 interacts with ASC are based on experiments that lack essential controls. For example, using cross linkers are known to generate false positive interactions. The authors did not use a negative control to rule out this possibility.

Authors: The interaction is probed by multiple different methods. Only one experiment used cross-linking of the proteins, the other experiments use a variety of other techniques. Together the experiments answer separate concerns about a particular protocol and so should be interpreted together. Additional controls have been added to experiments that probe the interaction between ZBTB16 and ASC (see changed Fig 6b) or are reanalysed (see updated

Fig 6a) and additional data has been added that alternatively measure this protein interaction (Fig 6d).

R1: Using over-expressed proteins also promotes false interactions, yet this concern is not remedied.

Authors: The interaction of the endogenous proteins is probed, thereby negating this concern (Figs 4a, 4b, 4e, 5e, 5f and 6d). Some of this data was in the initial submission, has been reanalysed or is new data in the resubmitted manuscript.

R1: The cross linker was used to show ASC oligomerization, however, ZBTB16 was not detected, hence, how do they suggest later that they interact?

Authors: ASC oligomerization and its interaction with ZBTB16 are separate events, which occur in different cellular compartments. We show that ZBTB16 interacts with ASC in the nucleus to promote its modification with SUMO1, while ASC oligomerizes in the cytosol. For this reason, ZBTB16 was not probed in the cross-linking experiment and so it is misleading to conclude that ZBTB16 was not detected.

R1: The authors do not put any molecular weight signs on any western, which makes it hard to interpret.

Authors: We have added molecular weight markers in the instances where these were omitted.

R1: All the confocal images are of very poor quality, the colocalization of red and green is not accurate due to overexpression of the two colors. In addition, to confirm colocalization, 3D and Z stack analyses are needed.

Authors: We have replaced the images with alternatives (Figs 5f and 6a) and have conducted a Z-stack analysis. We have also conducted an additional protocol (proximity ligation assay) to alternatively measure protein colocalizations (Fig 6d). The preceding data are now presented as supplementary data.

R1: The authors use the term ASC activity several times, however, it is not clear what is meant by its activity?

Authors: ASC functions as a bridging molecule that assembles the inflammasome. The text is amended to clarify the specific activity of ASC that is being discussed, as 'oligomerization' or its 'assembly of the inflammasome' if this isn't clear from the activity being probed -i.e., an association with NLRP3 or pro-Caspase 1 or the activation of Caspase 1 and pro-cytokine processing stemming from inflammasome formation.

R1: Using the production of IL-1B in the supernatant as a measure for inflammasome activity is incorrect, since it is well known now that IL-1B release into the supernatant relies on many factors that may be unrelated to the activation of the inflammasome. Accordingly, in some of their westerns, when IL-1B release is reduced, there is more accumulation inside the cell. Also, it usually accompanies LDH release which can be due to other caspases.

Authors: Inflammasome activity leads to the maturation and release of cytokines and so the measure does correctly capture inflammasome activity. The measures of the mature cytokine

by ELISA are accompanied by measures of the precursor and mature forms of IL-1 β and IL-18 by IB. Moreover, other protocols differently account for inflammasome activity. These include measures of the protein constituents and their interaction, ASC oligomerization and the visualization of the inflammasome speck, activation of Caspase 1 and recruitment of IL-1 β to the inflammasome complex as well as measures of LDH release. Notably, LDH release from apoptotic blebs after secondary necrosis occurs later than is probed in our experiment.

R1: They used the FLICA assay to measure caspase-1 activity, which is known to be non-specific.

Authors: This method uses a probe that encodes a defined Caspase-1 inhibitor (YVAD-fmk). This inhibitor binds with 10-fold lower specificity to other Caspases, such as Caspase 8 which might also be induced by the immune stimulus that is used in the experiment. Notably, the activity captured by the FLICA assay is congruent with all the other different measures of inflammasome activity and so is consistent with Caspase 1 activity. Additional data has been added to the amended manuscript that captured the recruitment of a recombinant IL-1 β molecule to the inflammasome that further supports the measure of Caspase 1 activity (Fig 4d).

R1: It is not clear why the reduction in IL-1B production in Nlrp3R258W/Zbtb16^{-/-} means that Zbtb16 acts downstream of Nlrp3.

Authors: The constitutive activity of the mutant Nlrp3 masks input from upstream regulators. Accordingly, suppression by ablating Zbtb16 puts it downstream of the hyperactive Nlrp3.

R1: Immunoblotting for immunoprecipitated proteins can suggest that 2 molecules are part of a complex, but it does not mean that they directly interact. Negative controls for all IP experiments were not included.

Authors: Additional control measures have been added in the revised manuscript where they were omitted. These are described fully below (in response to R3) but include the addition of a nonspecific IgG control (to Figs 5e and 6b) and a reverse IP control (Fig 6c).

As well as IP and IB, protein associations are probed by cross-linking and IB (Fig 4a), by immunofluorescence (Fig 4b, 5f, 6a and 7j) and with proximity methods by the proximity ligation assay (Fig 6d) and bimolecular complementation (Fig 6e-g and 7i). Crosslinking with DSS covalently binds amino acids that are within 12 angstroms. The proximity assays require a distance between proteins of less than 20-40 nm. Moreover, the biochemical consequences of the colocalization of ASC with ZBTB16, UBC9 and SUMO1 are demonstrated by the detection of the conjugation of SUMO1 to ASC. We also demonstrate the physiological consequences of the formation of the protein complex as altered inflammasome activity. Accordingly, there is an established functional outcome of colocalization.

R1: The authors suggest that ASC, ZBTB16 and SUMO1 interact but they did not present any data that actually demonstrate that SUMO1 SUMOylates ASC.

Authors: Figs 5a, b, c, d, e, 7a, b, e and Supplementary Fig 5a all demonstrate the conjugation of SUMO1 to ASC (i.e. ASC SUMOylation with SUMO1).

R1 Other concerns: Peritoneum pathogenesis was not scored

Authors: Peritoneum pathogenesis has now been scored and is shown in Fig 1b in the revised manuscript. A description of the method used has been added to the figure legend and materials and methods.

R1: Methods of statistical analysis of every panel should be added to the corresponding legends with the number of Ns.

Authors: The statistical methods used were given in the Material and Methods section. This information has now been repeated in each figure legend.

Reviewer #2 (Remarks to the Author):

The manuscript suggests interesting new functions and roles of ZBTB16/PLZF in inflammasome activity. The authors found ZBTB16 is important for promoting NLRP3 inflammasome activity by ZBTB16-deficient mouse experiments. ZBTB16 is known as a transcription factor, but the authors suggest that ZBTB16 is likely to have a different function in the inflammasome activity. The authors have shown that ZBTB16 binds and SUMOylates ASC (apoptosis-associated speck-like protein containing a CARD). The authors conclude that ZBTB16 positively regulates inflammasome activity via SUMOylation of ASC and the subsequent oligomerization of ASC. The data might contribute to the development of new therapeutic strategies to target inflammasome.

Almost all of the experiments are excellent and the evidence for the conclusion is convincing. The paper warrant publication in this journal after addressing several comments listed below.

R2 Comments: (1) In the discussion section, the authors suggest that ZBTB16 acts as a SUMO E3 ligase. The suggestion is very interesting, but it is necessary to access the E3 ligase activity of ZBTB16 using recombinant proteins.

Authors: We have been unable to demonstrate ASC SUMOylate in vitro. Correct SUMOylation of a control substrate (RanGAP1) vindicates our protocol. We believe the issue stems from ASC aggregation during purification (see figure). We haven't been able to overcome this technical obstacle. Notably, ASC monomers interact via the same interface that we have established to regulate its SUMOylation (Fig 7). The revised manuscript makes clear that we have not established ZBTB16 functions as an E3 SUMO ligase and we include additional discussion of the activity.

Figure: Coomassie-stained proteins electrophoretically separated under non-denaturing conditions.

- 1. Nondenaturing protein Marker II (45-669 kDa).*
- 2. Bovine Serum Albumin (BSA) as a 66 kDa reference protein -1000 ng.*
- 3. BSA -500 ng.*
- 5. ASC (22 kDa) purified as a dimer and tetramer.*

(2) In Fig 4a, the blot against ASC is unclear. It is better to re-examine this experiment.

Authors: We repeated the experiment and replaced the panel in Fig 4a.

(3) In Fig 5h, the signal of ASC in WT is weak. Is it difficult to detect SUMOylated-ASC?

Authors: In response to another Reviewer, we repeated this experiment using a different immune-enrichment protocol that has improved the recovery of ASC. This data replaces the former figure in the revised manuscript (Fig 5e).

It is a reported curiosity that SUMOylation induces its effect despite a low stoichiometry of modified substrates (although there are exceptions such as RanGAP1). This has been put down to turnover by SUMO proteases and/or cumulative downstream effects. This later concept envisages that SUMOylation drives the substrate into a functional complex or a subcellular compartment where the protein resides even after deconjugation. This is likely to be effective in responses where a substrate engages synchronous and concerted action by multiple components, such as in the assembly of immune signalling complexes.

(4) Recent papers reported that ZBTB16 can be degraded by several drugs like thalidomide derivatives (Matyskiela et al. ACS Chem Biol 2020, DOI: 10.1021/acscchembio.0c00674, Yamanaka et al. EMBO J 2021, DOI: 10.1021/acscchembio.0c00674, Shimizu et al. Commun Biol 2021, DOI: 10.1038/s42003-021-02801-y.). It might be better to discuss the therapeutic approach against pathogenic inflammasome activity by referring to these papers.

Authors: Additional discussion about the therapeutic opportunities that stem from our discovery has been added. We have also included additional data that demonstrates the effect of an inhibitor of SUMOylation on ASC modification and oligomerization (Fig 5b and 7i).

Reviewer #3 (Remarks to the Author): In the manuscript the authors claim that ASC activation and oligomerization is controlled by ZBTB16-mediated SUMOylation. However, the data do not support this conclusion. Thus, although the data demonstrate the SUMOylation of ASC, its relationship with protein oligomerization or its regulation by ZBTB16 is not probed.

Authors: Figs 1a-e, 2a-g, 3a-d, 4a-d demonstrate ZBTB16-dependent regulation of ASC-dependent inflammasome activity. Figs 4e (previously 4d), 5f (previously 5i) with new data shown in 6d and 6g demonstrate that ZBTB16 promotes the association between ASC and SUMO1, as well as other inflammasome proteins. Figs 5d, 5e, 6d and 7e demonstrate ZBTB16 promotes the SUMOylation of ASC. Fig 4a and b identify ZBTB16 promotes ASC oligomerisation. New data in Fig 7g-i demonstrate SUMO-dependent effects on ASC oligomerization. Together these data establish the ZBTB16-dependent SUMOylation of ASC, its subsequent oligomerization and the assembly of the inflammasomes to enhance inflammatory activity.

R3: The lack of molecular weight markers in many experiments as well as the lack of essential controls makes it difficult to interpret the results. There are multiple error in both the text and figures.

Authors: Markers and controls have been added where they were missing, and the text has been reviewed by an editor.

R3: The authors claim that ZBTB16 modulates ASC oligomerization; however, a reduction in the intensity of all the bands detected with the anti-ASC antibody (labelled as monomer, dimer and oligomer) is observed in Zbtb16^{-/-} cells (Fig 4a).

Authors: The levels of monomeric ASC (soluble) are equivalent in the WT and Zbtb16^{-/-} cells by IB. However, the extent of oligomeric ASC (insoluble) was decreased, consistent with reduced ASC oligomerization. The lysates from the experiment were re-probed to improve the quality of the IB. This data replaces the former Fig 4a, although the findings are not altered.

R3: Moreover, why the total amount of ASC protein (insoluble and soluble fractions) in WT cells treated with LPS and Nig is clearly higher than in LPS and Nig treated Zbtb16^{-/-} cells or in untreated cells? In panel d no differences are observed in the levels of ASC between WT and Zbtb16^{-/-} cells.

Authors: There were equivalent levels of soluble ASC in the figure initially presented. There remain equivalent levels in the revised Fig 4a. Levels of the insoluble fraction are lower in the Zbtb16 null cell because ASC oligomerization is reduced.

R3: Figure 4b. It is not clear from the pictures if there is less ASC speckles in Zbtb16^{-/-} cells or if the levels of the ASC are lower in these cells.

Authors: The levels of ASC are equivalent in the WT and Zbtb16^{-/-} cells. A quantification of ASC in this experiment has been added as Supplementary Fig 4. The levels of Asc are also shown to be equivalent in Figs 2c, 2d, 4e (previously 4d), repeated data 5e and 5f (previously 5h) and Supplementary Figs 5b and c.

R3: Figure 5a - It is not clear to me that the ASC-ZBTB16 co-localization takes place in the nucleolus as the authors say. What happens in untransfected cells? Does ZBTB16 overexpression induce ASC translocation?

Authors: We had not rigorously identified the nucleolus and so have removed this term. We now merely state that the proteins are in the nucleus. We quantitated the degree of co-localisation of ASC and ZBTB16 in z-stacks of the confocal data to better assess their location. This is shown as a reformatted Fig 6a in the revised manuscript. In addition, we add new data that alternatively measures the colocalization of endogenous ASC and ZBTB16 in untransfected cells by proximity ligation assay (Fig 6d in the revised manuscript).

R3: Why the subcellular distribution of ASC in Figure 5a and 5i is totally different from the one shown in Figure 7i?

Authors: The difference is due to enhanced detection of the overexpressed ASC constructs in the ASC^{-/-} BMDM in Fig 7 compared to endogenous Asc in Fig 5a and i. Notably, the data in Fig 5a and 5i have been replaced with better-quality images (labelled 6a and 5f).

R3: What happens with the ASC-ZBTB16 or ASC-SUMO co-localization in WT and ZBTB16^{-/-} cells after LPS+Nig treatment?

Authors: We replaced the former Fig 5i with 5f to show the effect of stimulation with LPS+nigericin. This shows an increased association between ASC and SUMO1 upon immune stimulation. Moreover, new data confirm the increased association between ASC and SUMO1, as well as with Zbtb16 and Nlrp3 upon stimulation (Fig 6d in the revised manuscript). Fig 7k (previously 7i) also shows ASC accumulates in the cytosol upon stimulation with LPS and aggregates as a cytosolic speck when co-stimulated with nigericin.

R3: Figure 5b - Why immunoprecipitated HA-ASC protein is detected as a double band? The experiments shown in figures 5b and 5c lack essential controls (immunoprecipitation using a control antibody)

Authors: Fig 5b & c had alternated the use of the anti-Flag and anti-HA antibodies for immune enrichment to control for non-specificity antibody affinity. However, this experiment has been repeated to, alternatively, include an IgG antibody as a nonspecific control in the revised manuscript (now Fig 6b and c). ASC appears as a single band in these IBs.

(Figure 5d) As a results of this experiment the authors conclude that ASC is mainly modified by SUMO1. However, the levels of unmodified Flag-ASC protein detected in the pull-down extracts is lower in those cells transfected with SUMO2 or SUMO3. Therefore, the absence of bands corresponding to SUMO2 or SUMO3 modified ASC protein in the blot could be due to a lower level of protein in those lanes.

Authors: The IB demonstrates that the levels of ASC are equivalent. The unmodified ASC is pulled down with the nickel resin either by an association with the HIS-SUMO without conjugation or by binding non-specifically to the resin. As it is not SUMOylated it is irrelevant to measures of SUMOylation. Re-extracted and re-probed the immune-enriched lysates from this experiment. This shows there is some variance in the levels of the unmodified ASC that is recovered, but there remains an absence of ASC modified by SUMO2 and SUMO3, thereby reconfirming the result. We have replaced the former figure panel with this replicated IB in the revised manuscript, although the finding is unchanged (Fig 5a).

R3: Figure 5e - In addition, analysis of the protein extracts after pull down using anti-Histidine antibody is also essential to interpret the data.

Authors: The IB with the anti-His antibody has been added to the figure. In addition to this experiment that supported ASC SUMOylation by demonstrating the removal of the signal by co-expressing the SENP1 SUMO protease (now coded Fig 5c), we include additional data that shows pharmacological ablation of the signal with a SUMO inhibitor (2-D08) (Fig 5b).

R3: Figure 5f & 5h- analysis of the immunoprecipitated proteins using anti-SUMO1 antibody is also essential evaluate the data.

Authors: These data showed the level of ASC co-immune enriched with an anti-SUMO1 antibody. We differently replicated this experiment by immune-enriching ASC from macrophages and then probing with an anti-SUMO1 antibody. These IBs, which include a probe with an anti-SUMO1 antibody, are presented in the revised manuscript (Fig 5e).

R3: Figure 5g - analysis of both input and Ni-NTA pulldown extracts with anti-Histidine antibody is also essential to interpret the data.

Authors: The IB with the anti-His antibody has been added (relabelled as 5d in the revised manuscript).

R3: Figure 5h - Why is the pattern of the ACS protein detected in the SUMO1 immunoprecipitated extract in this figure totally different from the one shown in figure 5f?

Authors: The IBs focused on different molecular weight proteins (5d<70; 5f<200 kDa). These two panels have been replaced by Fig 5e in the revised manuscript. Additionally, it should be appreciated that the apparent molecular weight of ASC is not solely due to the conjugation of SUMO1 but will capture other post-translational modifications.

R3: Figure 5i - It seems that the SUMO1 signal in the ZBTB16^{-/-} cells is lower than in the WT cells (*and later*) The WB analysis to compare the levels of SUMO1 present in both cells included in supplementary information should show both free and conjugated SUMO1 protein. The presence of lower levels of SUMO1 in ZBTB16^{-/-} cells could explain a lower co-localization with ASC.

Authors: The levels of SUMO1 are equivalent in the WT and Zbtb16^{-/-} cells. The previous data shown in Fig 5i (now 5f) has been changed to show the effect of an immune stimulus. Quantification of the relative levels of SUMO1 from this experiment is shown in Supplementary Fig 5b. Other data confirm equivalent levels of free and conjugated SUMO1 in WT and Zbtb16^{-/-} cells (Figs 5e and Supplementary Fig 5c).

R3: Does ZBTB16 promote the SUMOylation of ASC in vitro?

Authors: We have been unable to get the in vitro assay to work. As we responded to R2 above, we believe this is due to ASC aggregation during purification, which interferes with its association with the SUMO ligase/SUMO1. We have included a discussion on the lack of certainty of the mechanism by which ZBTB16 increases ASC SUMOylation in the revised manuscript.

This additional discussion also encompasses a recent report that identifies that disruption of protein binding at the ASC PYD, which we identify is SUMOylated, enhances inflammasome activity (doi.org/10.1038/s41420-023-01438-6). Related to this, we have included additional data that probes the effect of SUMOylation on ASC oligomerization (Fig 7g-i).

We also discuss another report that identified that the modification of the innate immune adaptor, Mitochondrial antiviral signalling protein, with SUMO potentiates the antiviral response as a possible exemplification of how SUMOylation may promote the assembly of immune complexes (doi.org/10.1038/s41594-023-00988-8).

R3: Figure 6a - The number of fluorescent dots is different in each panel. Moreover, cells transfected with ASC-ASC plasmids only show one dot.

Authors: The pattern of fluorescence is as expected as ASC homo-oligomerizes as a single speck in the cytoplasm (see Fig 4b and 7i-j) while it forms multiple heterologous interactions with UBC9, SUMO1 and ZBTB16 in the nucleus. New data that alternatively probes the proximity of these proteins by proximity assay replicates this pattern (Fig 6d).

R3: Figure 6 - How significant is this data? Western blot analysis on lysates of transfected cells in each BiFC experiment is required to determine if both fusion proteins are expressed equivalently.

Authors: It is difficult to achieve equivalent molar ratios of the different molecular weight proteins. Measures of the levels of protein expression by IB are crude and are not sufficiently sensitive to normalize expression to differences in the fluorescent intensity. To avoid over-analysis, we merely use Venus fluorescence to confirm an association between the different constructs. Accordingly, statistical measures of the different fluorescent signals are not calculated. Fig 6g (previously 6c) is an exception as the same constructs are used and so can be compared (the statistical comparison is shown). The expression of the different constructs is confirmed by the generation of a fluorescent signal in at least one of the protein pairings.

R3: Figure 6 - analysis of the subcellular localization of each fusion protein using antibodies specific for each fusion protein is also essential to evaluate whether the fusion of the fluorescent fragment alters the subcellular localization of the protein of interest.

Authors: The subcellular location of the tagged proteins is visualized by Venus fluorescence, as apparent in the micrographs (Fig 6e). The cellular location of the collocated, split-Venus proteins is equivalent to that detected in the other experiments using fluorescent tagging, immunofluorescence and proximity ligation assay.

R3: Figure 6b - It is not clear how the fluorescence was quantified.

Authors: The quantitation method is described in the material and methods. It is simply the total Venus fluorescence averaged across multiple images after subtracting background fluorescence.

R3: Do ASC and SUMO1c co-localize?

Authors: This wasn't measured as, unlike ZBTB16, ASC is not predicted to encode a SUMO-interacting motif.

R3: Figure 7b-d & f - analysis of the protein extracts after pull down using anti-Histidine antibody is also essential to interpret the data. WB of the whole cells extracts with anti-Flag antibody is also required.

Authors: These requested IB have been added to the figure (Now Fig 7a, 7b, 7e and 7f).

R3: Figure 7b - The authors conclude that mutation of some specific lysine residues reduces ASC SUMOylation. However, the intensity of the bands corresponding to the unmodified mutant proteins is less than that of the unmodified WT protein. Therefore, a reduction in the intensity of the SUMOylated bands could be due to a lower level of protein in the Ni-NTA pull down extracts.

Authors: The expression of the different ASC constructs is shown to be equivalent, so does not account for the difference. As described above, the unmodified (low molecular weight) ASC that is copurified with the nickel resin isn't SUMOylated, and so, is not relevant to measures of SUMOylation.

R3: Figure 7c - Mutation of K109 seems to reduce SUMOylation. Why haven't the authors made a mutant by combining this lysine with the mutated ones in Figure 7b?

Authors: As was shown, mutation of the residues around K21 disrupts ZBTB16's association with ASC and so reduced all SUMOylation of ASC (Fig 7a, d, e and Supplementary Fig 7d). Accordingly, the mutations are redundant. However, new data is added in the revised manuscript showing that the combined mutant (K21R + K109R) is equivalent to the single mutations (K21R or K109R) in its effect on the formation of the ASC speck (Fig 7j).

R3: Figure 7d - Several panels in Figure 5 show also WB analysis of Flag-ASC protein in His-SUMO1 pull-down extracts. In all those panels, bands corresponding to both the unmodified and modified ASC protein can be observed. Why is only one band detected in this blot? The authors state that the 4KR has lost the SUMOylation. However, SUMOylation of the 4KR mutant is clearly observed.

Authors: As discussed above the low molecular weight ASC is not SUMOylated and so is irrelevant for analysis of this modification. The extent of its recovery in this procedure is a distraction. As Fig 7d merely identified a secondary effect of the Flag tag, which was pertinent to interpreting the experiment, it has been presented as supplementary data in the revised manuscript (Supplementary Fig 7d). We changed the text to state that the SUMOylation of the 4KR mutant is 'reduced', rather than 'lost' as is evident in the figure.

R3: Figure 7g - Is this IP carried out in denaturing or non-denaturing conditions? If it has been carried out in non-denaturing conditions, the bands detected with anti-ubiquitin antibody could correspond to the ubiquitinated proteins that interact with ASC instead of the ubiquitinated-ASC protein.

Authors: The IP used denaturing conditions. It isn't clear why this constitutes a concern. ASC is enriched in the experiment and so will represent the major component of the IP, but all immune-enriched proteins are electrophoretically separated and probed and so can be compared. The comparison shows no difference in ubiquitinated ASC or any possible co-enriched proteins.

R3: Figure 7h - Evaluation of the oligomerization state of the ASC mutants is essential to understand the involvement of SUMO on the regulation of ASC oligomerization.

Authors: The oligomerization of ASC was assessed by measures of its soluble and insoluble forms, detection of ASC specs in the cytosol and its association with inflammasome proteins and subsequent activation of the inflammasome, which all require ASC oligomerization. Additional data demonstrating that SUMOylation controls the homo-oligomerization of ASC has been added (Fig 7g-i in the revised manuscript).

R3: Figure 7i - In those cells treated with LPD and Nig, the signal of the 4KR mutant is clearly lower. Is the protein being degraded?

Authors: Nigericin induces cell death (by pyroptosis), thereby reducing the cells and accompanying proteins levels.

R3: Figure 8 - The authors propose that ASC is modified by SUMO in the PML-NBs. However, in the manuscript the authors state that PML and ZBTB16 co-localize inside the nucleolus?

Authors: As we hadn't rigorously defined the subnuclear compartment we changed our use of PML-NBs and nucleolus to 'nucleus' and 'subnuclear structures' to describe these compartments in the context of ASC and ZBTB16s.

REVIEWERS' COMMENTS

Reviewer #1 (Remarks to the Author):

the authors addressed all my concerns

Reviewer #2 (Remarks to the Author):

The manuscript has been substantially improved, I feel that the authors conducted an excellent job. I think this paper warrants publication if no other reviewers have any objections.

Reviewer #3 (Remarks to the Author):

The authors have addressed all the major concerns. There are some spelling mistakes (for example, fig 4a, "oligmer")

REVIEWERS' COMMENTS (26th, September, 2023)

Reviewer #1 (Remarks to the Author):

R1: the authors addressed all my concerns

Authors: We are very grateful for your recognition of our work

Reviewer #2 (Remarks to the Author):

R2: The manuscript has been substantially improved, I feel that the authors conducted an excellent job. I think this paper warrants publication if no other reviewers have any objections.

Authors: We are very grateful for your recognition of our work

Reviewer #3 (Remarks to the Author):

R3: The authors have addressed all the major concerns. There are some spelling mistakes (for example, fig 4a, "oligmer")

Authors: edited to the "oligomer" in the Fig4a

We are very grateful for your recognition of our work